# Quantum vertex model for reversible classical computing

C. Chamon[1], E.R. Mucciolo[2], A.E. Ruckenstein[1] & Z.-C. Yang[1]

Mappings of classical computation onto statistical mechanics models have led to remarkable successes in addressing some complex computational problems. However, such mappings display thermodynamic phase transitions that may prevent reaching solution even for easy problems known to be solvable in polynomial time. Here we map universal reversible classical computations onto a planar vertex model that exhibits no bulk classical thermodynamic phase transition, independent of the computational circuit. Within our approach the solution of the computation is encoded in the ground state of the vertex model and its complexity is reflected in the dynamics of the relaxation of the system to its ground state. We use thermal annealing with and without 'learning' to explore typical computational problems. We also construct a mapping of the vertex model into the Chimera architecture of the D-Wave machine, initiating an approach to reversible classical computation based on state-of-the-art implementations of quantum annealing.

[1] Physics Department, Boston University, 590 Commonwealth Ave., Boston, Massachusetts 02215, USA. [2] Department of Physics, University of Central Florida, 4111 Libra Drive, Orlando, Florida 32816, USA. Correspondence and requests for materials should be addressed to C.C. (email: chamon@bu.edu).

Throughout the past few decades, problems of computer science have become subjects of intense interest to theoretical physicists as paradigms of complex systems that could benefit from theoretical approaches and insights inspired by statistical physics. These include neural networks, Boltzmann machines and deep learning, compressed sensing, satisfiability problems and a host of other approaches to data mining and machine learning[1–4]. The interest in the constraints on computation and information processing placed by physical laws is even older and dates to work by Landauer and Bennett[5–7]. One of the holy grails at the interface between physics and computer science is the physical realization of a large-scale quantum computer in which the processing of information makes use of quantum-mechanical concepts such as superposition and entanglement[8,9]. However, building a quantum computer remains a challenging task because of the practical difficulty associated with maintaining coherence over the duration of the computation.

This paper aims at bringing a new class of problems to the physics–computer science interface by introducing a two-dimensional (2D) representation of a generic reversible classical computation, the result of which is encoded in the ground state of a statistical mechanics vertex model with appropriate boundary conditions. The vertex model is defined in terms of Boolean variables (or spins degrees of freedom) placed on the bonds or links of an anisotropic 2D lattice with vertices representing logic gates. The corresponding gate constraints are implemented through short-ranged one- and two-body interactions involving the spins of the vertex (as we show, this construction can be realized in physical programmable machines, such as the D-Wave machine.) One direction of the lattice represents 'computational (rather than real) time', as introduced by Feynman in the history representation of quantum computation[10], but here used for classical reversible circuits. The two boundaries of the lattice transverse to the 'time' direction contain the input and output bits of the computation. It is important to stress that we are not limiting ourselves to forward computations with fixed inputs. More interesting are problems in which only partial information about both inputs and outputs is known. In that case, reaching the ground state requires flow of information both forwards and backwards across the lattice, processes that are naturally built into our approach.

The idea of encoding classical computation in the ground state of a many body spin model was introduced earlier for irreversible computation in refs 11–13. Here we focus on reversible rather than irreversible computation to address problems with both fixed-input and mixed-boundary conditions on inputs and outputs, as explained above. Mapping onto a regular 2D lattice as opposed to an arbitrary graph allows us to use intuitive ideas from equilibrium and non-equilibrium statistical mechanics, especially of classical and quantum phase transitions. Also, while in ref. 12 an error correction scheme was required to implement fault tolerant computation, in our approach accurate computation without error correction is possible below moderate temperatures that scale only as the inverse of the logarithm of the system size, a consequence of the exponential scaling of the static correlation length with inverse temperature (see below).

Most importantly, the mapping proposed here defines statistical mechanics vertex models that, irrespective of the computation they represent, display no bulk thermodynamic transition down to zero temperature. Thus our work emphasizes that the dynamics of relaxation to the ground state rather than the thermodynamics of the model is essential for understanding the complexity of ground-state computation.

The absence of a thermodynamical phase transition removes an obvious impediment to reaching the ground state of the vertex model. For instance, a suboptimal mapping from a computational problem into a physical system may place the solution within a glassy phase, even in the case of easy computational problems. The mapping of XORSAT (a problem in P) into a diluted $p$-spin model is such an example[14]. The fact that our vertex model is free of thermodynamic transitions does not mean that the ground state can be reached easily. This remains true even for problems with unique solutions which are encoded by vertex models with unique ground states. Such problems are in the complexity class UNIQUE-SAT, which under randomized reduction is as hard as SAT[15]. Hence, even in the absence of a thermodynamic transition finding the unique ground state of vertex models encoding problems with a single solution is a problem in NP-complete[16–18]. Of course, this does not mean that one cannot benefit from speed-ups allowed by either physics inspired heuristics or by special-purpose physical hardware, such as quantum annealers.

This paper focuses on the study of vertex-model representations of random circuits for which the complexity of the computation is reflected in the concentration of TOFFOLI gates, the length of the input and output boundaries $L$ and the depth of the circuit $W$. We concentrate on computational problems with a single solution—or problems for which one can discern among an $\mathcal{O}(1)$ number of solutions with a small overhead—a class of problems that encompass factoring of semi-primes, an important and nontrivial example that we shall explore in a future publication.

In our discussion of dynamics we deploy thermal annealing as well as introduce a more efficient 'annealing with learning' protocol. The latter translates into an algorithm for solving classical problems for which, as expected, forward computation from a fixed-input boundary reaches solution in a time linear in the depth of the computational circuit. Finally, we note that reaching the ground state of the vertex model could be accelerated by replacing classical annealing with quantum annealing[19–22]. While approaching computational problems through quantum annealing is left for future investigations, the current paper includes the formal derivation of the quantum version of the statistical mechanics model of reversible classical computation. This provides the background for an explicit mapping of our lattice model onto the Chimera architecture of the D-Wave machine, a development that points to the potential usefulness of the vertex model as a programming platform for special purpose quantum annealers.

## Results

**The vertex model for reversible classical computation.** Our starting point is the fact that any Boolean function can be implemented in terms of TOFFOLI gates, which are reversible logic gates with three inputs and three outputs. Starting from a circuit of TOFFOLI gates, our construction proceeds by first using SWAP gates to repeatedly swap distant bits in the input that are acted upon by particular gates of the circuit, until the operation of every gate is reduced to adjacent bits. The second step is to associate tiles with each of the gates, as shown in Fig. 1, where one should imagine placing input and output bits at the intersections of the tile surfaces with the horizontal lines, as described in detail in Methods section.

The tiles representing the gates can then be laid down side-by-side on a plane to implement the computational circuit, as shown in Fig. 2 for the example of the 'ripple-carry adder', which computes the carry bit that is 'rippled' to the next bit when adding two numbers[23]. (The 'ripple-carry adder' is the building block for more complicated circuits such as addition and multiplication.) As can be seen from this example, one may

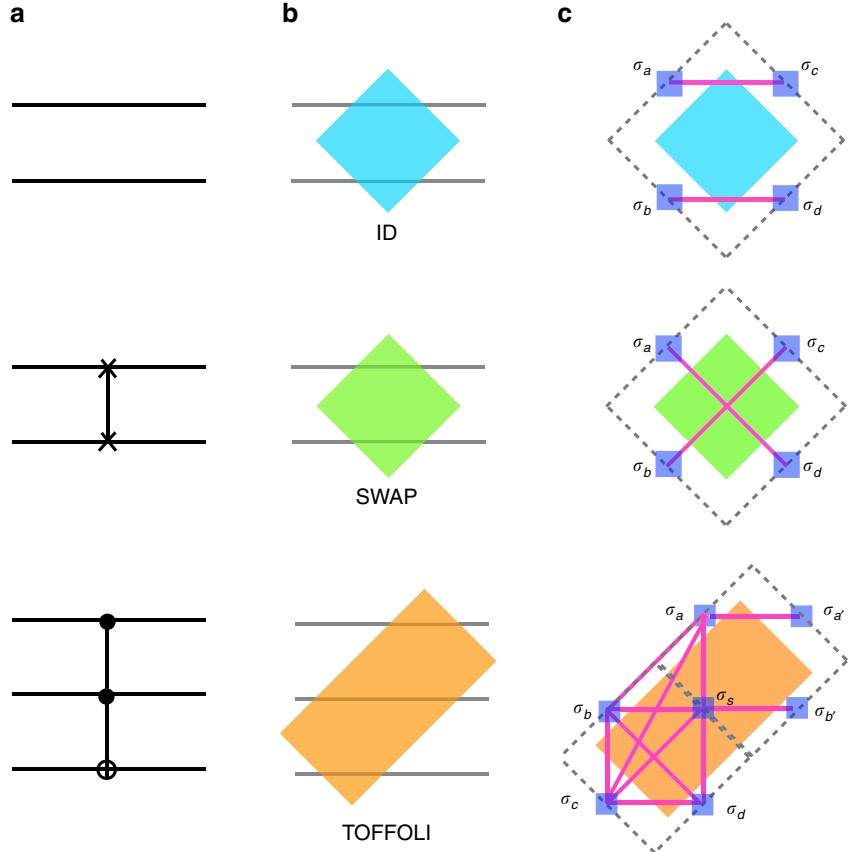

**Figure 1 | Tile representation of reversible computational gates.** (**a**,**b**) Elementary tiles representing the three computational gates for reversible circuits: ID (identity), SWAP and TOFFOLI. (**c**) construction of the Hamiltonians that encode the gate-satisfying states in the ground-state manifolds. Spins are placed on the boundary of the tiles. For the TOFFOLI gate, an ancilla spin is placed in the centre of the rectangular tile. Couplings needed in the Hamiltonians for the three different gates (tiles) are indicated by purple lines connecting two spins. The dashed line denotes the boundary of the tile.

also need to include the Identity (ID) gate in addition to the TOFFOLI and SWAP gates to represent particular logic circuits via tiling. Implied in the figure is that common boundaries of adjacent tiles contain a pair of 'twin' bits (one on each tile) whose values must coincide. The derivations of spin Hamiltonians implementing the truth tables of individual tiles, the short range inter-tile Hamiltonian enforcing the consistency between bits of neighbouring tiles, and the boundary conditions specifying inputs and outputs are presented in Methods section.

The final step of our mapping, also detailed in Methods section, is to construct a vertex model on a tilted square lattice, with each vertex representing either a TOFFOLI gate or four possible rectangular tiles obtained by combining square ID and SWAP tiles (ID–ID, ID–SWAP, SWAP–ID, SWAP–SWAP), as shown in Fig. 2. This construction can always be done by an appropriate retiling of the circuit so that each rectangular tile has four neighbours (hence the square lattice). There are six Boolean (or spin) variables associated to each vertex: two on each of the two double bonds and one on each of the two single bonds tied to a vertex. In deriving the vertex model we work in the limit in which the spin coupling defining the gate Hamiltonians, $J \to \infty$ (see Methods section), in which case all gate truth tables are satisfied exactly. Consequently, each vertex can be in one of $r = 2^3 = 8$ states. Three of the spins are inputs, and we use the state $q$ of the vertex, where $q = 0, 1, \ldots, 7$, to read-off the inputs in binary (which are uniquely related to the spin): $x_a^{\mathrm{IN}} = \mathrm{bit}[a, q]$, $a = 1, 2, 3$ for the three bits of the number $q$. The output bits are the bits of the 3-bit number $G(q)$, where $G$ is the gate function: $x_a^{\mathrm{OUT}} = \mathrm{bit}[a, G(q)]$, $a = 1, 2, 3$. The energy cost for two adjacent

gates that are incompatible with each other is determined by the ferromagnetic coupling $K$.

The resulting vertex model Hamiltonian can be written as

$$
\begin{aligned}
\hat{H} = & \sum_{\langle ss' \rangle} \sum_{q_s, q_{s'}} K_{q_s, q_{s'}}^{g_s, g_{s'}} \, |q_s q_{s'}\rangle \langle q_s q_{s'}| \\
& + \sum_{s \in \text{boundary}} \sum_{q_s} h_{q_s} \, |q_s\rangle \langle q_s| \\
& + \sum_s \sum_{q_s, q_s'} \Delta_{q_s, q_s'} \, |q_s\rangle \langle q_s'| \,,
\end{aligned}
\tag{1}
$$

where $K_{q_s, q_{s'}}^{g_s, g_{s'}}$ encodes the energy cost for mismatched nearest-neighbour vertices (the energies, with scale set by $K$, depend on the state of the vertices $q_s$ and $q_{s'}$, as well as on the types of gates $g_s$ and $g_{s'}$ present at neighbouring vertices $s, s'$—an explicit example is given in Supplementary Note 2); $h_{q_s}$ encodes the boundary conditions, which we associate directly with the vertex rather than with the input or output bits of a gate (since the relationship is one-to-one); and finally, the transition matrix elements $\Delta_{q_s, q_s'}$ between the states within a vertex $s$. All these couplings can be determined given a computational circuit and the boundary conditions. The quantum term $\Delta_{q_s, q_s'}$ can be designed from the internal couplings within the tiles. For simplicity, one should consider the case, $\Delta_{q_s, q_s'} = \Delta$ for all $q_s, q_{s'}$, which then represents the eight-state counterpart of a transverse field.

The vertex model defined by equation (1) is the starting point for all the subsequent discussions of this paper. For example, a quantum annealing protocol for solving a factoring problem would start with $K \ll \Delta$, where the ground state is a superposition

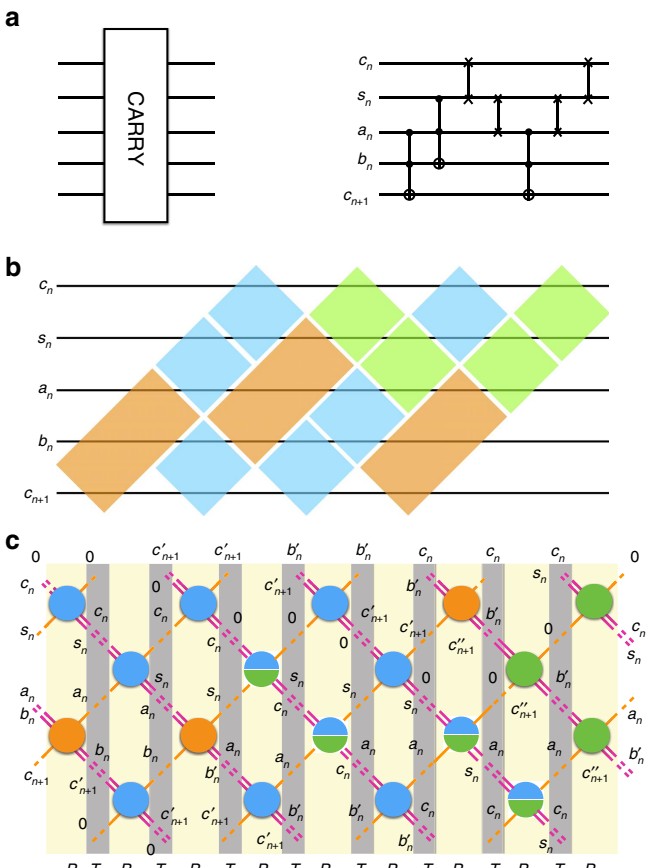

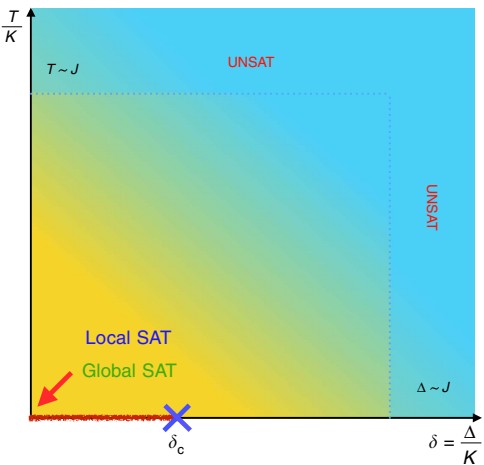

**Figure 3 | Phase diagram of the vertex model.** Our exact calculation of the partition function shows that there is no phase transition along the classical path ($\delta = 0$). We argue that there should be a quantum phase transition for some critical $\delta_c$.

Moreover, along the 'quantum' axis $T = 0$ the vertex model must encounter a zero-temperature quantum phase transition at a finite value of $\delta$. This follows from considering trivial classical circuits with no TOFFOLI gates in which case an $L \times W$ vertex model is equivalent to $3L$ decoupled Ising chains of size $W$ in a transverse magnetic field. Just as in the one-dimensional Ising model in a transverse field, in the limit of no TOFFOLI gates one expects a zero-temperature second-order quantum phase transition at $\delta_c = 1$. The addition of TOFFOLI gates complicates the analysis, but on physical grounds we expect that the phase transition cannot simply disappear but rather change character instead, possibly from second order to first order. This could be the case if the no-TOFFOLI critical point happens to be an endpoint of a phase boundary in the $\delta$–$x_T$ plane, where $x_T$ is the concentration of TOFFOLI gates. Determining the order of the transition for the vertex model describing a generic computation is a difficult problem, which we expect to address via quantum Monte-Carlo simulations in a future publication.

**Thermal annealing of the classical vertex model**. Here we study the dynamics of relaxation to the ground state as a function of the size and depth of the computation via thermal annealing[24]. This proceeds by cooling the system from a high temperature of order $K$ down to zero temperature over a total time duration, $\tau$, according to the ramp protocol, $T(t) = K(1 - t/\tau)$.

The dynamics is extracted by following an order parameter $m$ that measures the overlap of the final state $\{q^{\text{final}}\}$ reached at $t = \tau$ with the reference (solution) state $\{q^{\text{sol}}\}$:

$$m = \frac{8}{7}\left[\frac{1}{LW}\sum_s \delta_{q_s^{\text{final}},\, q_s^{\text{sol}}} - \frac{1}{8}\right]. \tag{3}$$

(Below we explain in detail how a unique solution state $\{q^{\text{sol}}\}$ is obtained.) Notice that the order parameter reaches $m = 1$ when the final state agrees with the solution, and $m = 0$ if the state is random, in which case it agrees with the solution by chance in 1/8th of the sites. We remark that the 'solution overlap' is a much better indicator of the evolution towards solution than the total energy. This is because a single vertex flip into an incorrect state in the middle of the circuit may cost little energy but it throws other vertices into a completely different state from the correct one.

**Figure 2 | Tile and vertex model representation of the ripple-carry adder.** (**a**) The ripple-carry adder which computes the carry bit that is 'rippled' to the next bit. We add one additional control line $s_n$ and set it to 1 to implement the original CNOT gate with a TOFFOLI gate. (**b**) The ripple-carry adder implemented on the tile lattice, with different gates depicted in different colours: blue tile: ID; green tile: SWAP; gold tile: TOFFOLI. Spins between adjacent tiles are forced to be equal by the ferromagnetic 'grout' coupling $K$. (**c**) The ripple-carry adder mapped to a vertex model with periodic boundary condition in the transverse direction. After each column of gate (vertex) operation, bit states are labelled at each bond. Light yellow and grey stripes represent the $P$ and $T$ matrices used in the transfer matrix calculation of the partition function.

of all locally satisfied gates independent of one another, and end with $K \gg \Delta$, with the ground state in which each tile satisfies the gate constraint and also passes and receives the right information to and from its neighbours.

**The quantum vertex model phase diagram**. Figure 3 shows our conjectured equilibrium phase diagram of the vertex model described by the Hamiltonian in equation (1). For $T, \Delta \ll J$, the local gate constraints are satisfied, which we indicate by 'local SAT'. The solution of the computational problem resides at the origin ($T/K = \Delta/K = 0$), where all the gates are locally satisfied and globally consistent, which we indicate by 'global SAT'. In Methods section we show explicitly that along the classical axis, $\delta = \Delta/K = 0$, the vertex model displays no finite temperature bulk thermodynamic transition irrespective of the computational circuit it represents. In particular, the resulting bulk thermodynamic behaviour is always that of a paramagnet:

$$\beta F = -[3L(W-1)]\ln(2\cosh\beta K). \tag{2}$$

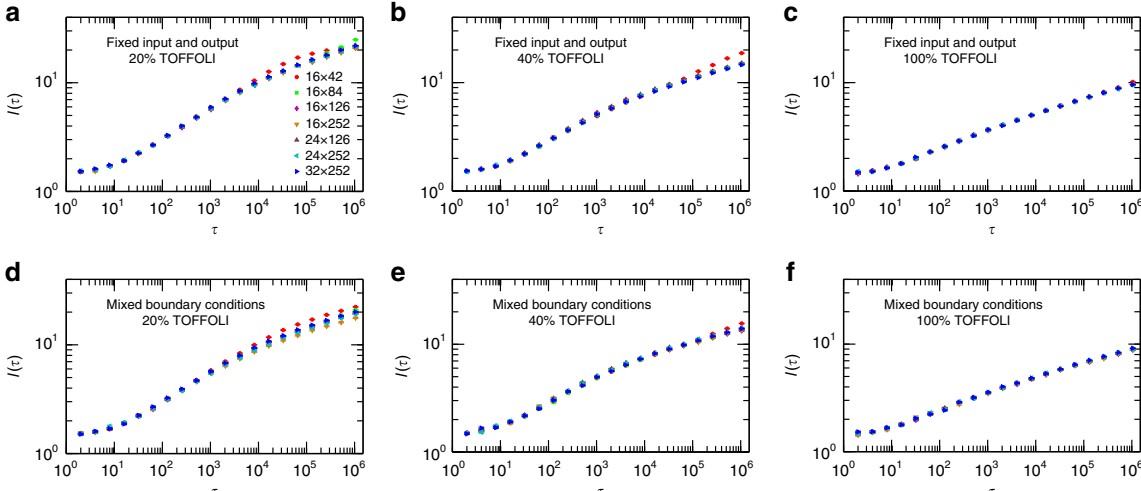

**Figure 4 | Scaling of the dynamical correlation length $\ell(\tau)$.** For all sizes and cases, 2,000 realizations of the boundary states were used. The data-point code used in **a** applies to all other panels. When not visible, the error bars are smaller than the size of the data points. (**a**–**c**) Fixed input and output; (**d**–**f**) mixed boundary conditions; (**a**,**d**) 20% TOFFOLI; (**b**,**e**) 40% TOFFOLI; and (**c**,**f**) 100% TOFFOLI. For systems with the smallest depth $W$ studied, $16 \times 42$, and for circuits with few TOFFOLI gates (20%) and fixed-input and output boundary conditions, $\ell(\tau)$ tends to saturate, indicating that complete solutions have been reached. Notice that the functional form of the scaling does not depend on the boundary conditions, and depends solely on the concentration of TOFFOLI gates.

The details of the numerical Metropolis simulations are presented in Methods section. Our results are represented in the form inspired by the dynamic scaling theory of ref. 25 that builds on the Kibble–Zurek mechanism[26,27], namely:

$$\ell(\tau) = \langle m \rangle(\tau) \, WL/L_\partial, \qquad (4)$$

which defines a dynamical correlation length, $\ell(\tau)$. $L_\partial$ is the number of pinned vertices on both boundaries (see Methods section). To motivate equation (4) we note that the domain of satisfied gates that contribute to $\langle m \rangle(\tau)$, the fraction of gates that reach their correct states at time $\tau$, grows from the pinned states at the boundaries, and covers an area $L_\partial \times \ell(\tau)$. Thus $\ell(\tau)$ describes the growth of correlated regions of satisfied gates that eventually connect the two boundaries of the circuit. (We note that recently, the Kibble–Zurek mechanism has been extended to include systems with zero-temperature order[28], the case relevant to the current discussion.)

We note that at any temperature $T$ along the annealing path, the correlation length is $\ell_T(\tau) \leq \ell_T(\tau \to \infty) = \xi_T$, where $\xi_T \sim e^{K/2T}$ is the thermal correlation length in the paramagnetic state, and $K$ is a characteristic ferromagnetic interaction strength in our model. In thermal equilibrium all gate constraints defining the computational circuit would be satisfied once $\xi_T$ reaches the depth of the computation, $W$. Notice that the exponential dependence of $\xi_T$ on temperature implies that achieving the correct assignment of gates does not require very low temperatures on the scale of $K$ since $\xi_T \sim W$ already for temperatures below $T \sim K/\ln W$. However, reaching the solution to the computational problem is a dynamical process that cannot proceed to completion until the dynamic correlation length at the end of the annealing protocol, $\ell(\tau) = \ell_{T=0}(\tau)$, reaches $W$, allowing the input and output boundaries of the system that specify the computation to communicate.

In Fig. 4 we present the numerical results for fixed-input and output, and mixed boundary conditions, with different concentrations of TOFFOLI gates (see Methods section for details). Remarkably, we find that the curves for different system sizes $L$ and $W$ collapse very well when scaled as in equation (4). In addition, notice that for shorter circuit with fixed input and output and low concentration of TOFFOLI gates (20%), $\ell(\tau)$

begins to saturate for large enough $\tau$ (Fig. 4a). As shown more clearly in Fig. 5a, this saturation occurs when the dynamical correlation length $\ell(\tau)$ reaches $W/2$, where the growing domains of satisfied gates meet. Since in this case $L_\partial = 2L$, $\ell(\tau_s) \sim W/2$ corresponds to $\langle m \rangle(\tau_s) = 1$ establishing $\tau_s$ as the time-to-solution. For mixed boundary conditions, however, $\ell(\tau) \sim W/2$ initiates the communication between the two boundaries and establishes the system's capacity to 'learn' (see below) but is not sufficient for negotiating solution. Indeed, Fig. 5a shows that $\langle m \rangle(\tau)$ does not yet saturate when $\ell(\tau) \sim W/2$. As can be seen from Fig. 6 for computations with mixed boundary conditions, correlations must develop along the transverse direction (that is, parallel to the boundaries) before solution can be reached. In those cases it is this slower process that determines the time-to-solution and dominates the complexity of computations.

Finally, all non-trivial operations between input and output bits involve TOFFOLI gates, and it is thus expected that the increasing the concentration of these gates slows down the growth of correlations. This expectation is confirmed in Fig. 5b, where we show curves for the same system size with different concentrations of TOFFOLI gates. The case of no TOFFOLI gates is equivalent to $3L$ decoupled ferromagnetic Ising chains. In this case the dynamic correlation length behaves as $\ell(\tau) = \ell_0 \big[ (\tau/\tau_0)/\ln(\tau/\tau_0) \big]^{1/2}$ (with $\ell_0 = 1.42$ and $\tau_0 = 8.33$) as illustrated by the dashed line in Fig. 5b. This behaviour is in agreement with the exact result for the Kibble–Zurek dynamical scaling of the density of domain walls in a ferromagnetic Ising chain[29].

**Annealing with learning.** Simple thermal annealing is not necessarily an optimal way to reach the ground state. For example, in the case of forward computation, the time scale for the dynamical correlation length to grow to $\ell(\tau) \sim W/2$ (so as to reach solution) is slower than ballistic (or linear in $\tau$), as expected for deterministic forward computation. This can be already seen from the exactly solvable case with no TOFFOLI gates. Moreover, for single-solution problems with mixed boundary conditions the growth of correlations establishing communication between boundaries scales with the same form as in direct computation (Fig. 4). However, in that case negotiating solution requires the

establishment of much slower correlations along the boundaries, a process for which a 'vanilla' thermal annealing approach is extremely inefficient and would require unreasonably large computational resources.

These shortcomings are addressed by using a heuristic 'learning' protocol in which annealing proceeds through the

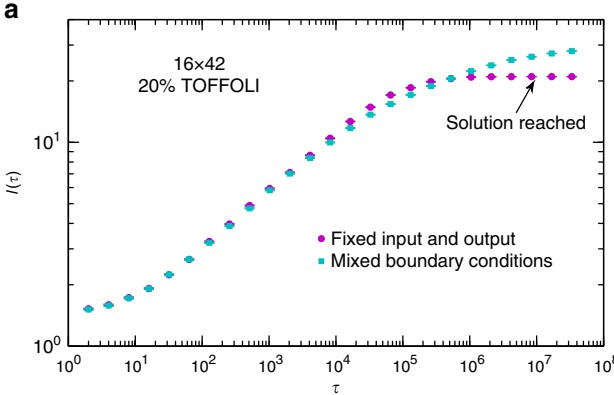

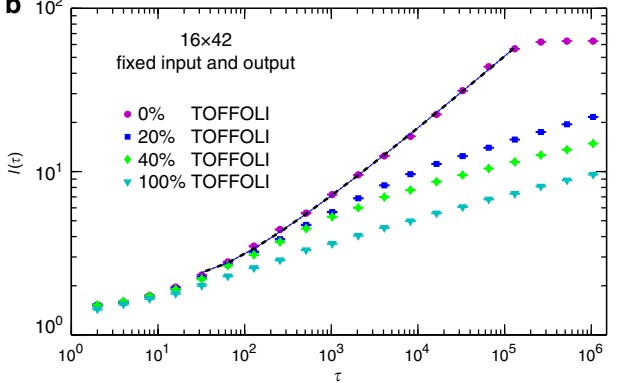

**Figure 5 | Effects of boundary conditions and TOFFOLI concentration.** (**a**) The saturation of the scaling curve for fixed-input and output boundary conditions with 20% TOFFOLI gates when $\ell(\tau) \sim W/2$. This indicates that the solutions have been reached, and is consistent with the domain growth picture. Notice that for mixed input and output boundary conditions, the curve does not saturate when $\ell(\tau) \sim W/2$. (**b**) The functional form of the scaling curves as a function of different TOFFOLI concentrations. The correlation grows more slowly as the concentration of TOFFOLI gates increases. The dashed line corresponds to the fitting $\ell(\tau) = \ell_0[(\tau/\tau_0)/\ln(\tau/\tau_0)]^{1/2}$, with $\ell_0 = 1.42$ and $\tau_0 = 8.33$.

following steps: (1) one starts by annealing $N_R$ identical replicas of a circuit over some time $\tau_a$, during which the correlation lengths grow beyond a few columns of gates such that the probability for assigning correct gates within that region, $p \sim \exp(-|x|/\xi) > 1/2$, within each replica; (2) one then assigns a specific identity to each gate (with $p > 1/2$) provided that a fraction of the $N_R$ replicas, greater than or equal to $\alpha$, agree on this assignment; (3) with the agreed upon gates frozen, the annealing process is independently applied again to each of the replicas allowing only gates not yet fixed to participate in the Metropolis algorithm; and finally, (4) the procedure is iterated until all gates are fixed, thus establishing the solution to the problem.

This protocol raises the question of how many replicas $N_{R\varepsilon}$ are needed to ensure that the learning algorithm reaches the correct result with a probability greater than $1 - \varepsilon$. In particular, how does $N_{R\varepsilon}$ depend on the system size $L \times W$ and the threshold $\alpha$? As we show in Supplementary Note 3, the number of replicas needed to ensure an error rate smaller than $\varepsilon$ is given by $N_{R\varepsilon} = \frac{\ln\left[\frac{2p-1}{p}\frac{\varepsilon}{LW}\right]}{\ln[2p^{1-\alpha}(1-p)^\alpha]}$, where $p > \frac{1}{2}$ is the probability of a correct gate assignment for one replica. Note that, for fixed $\alpha$ and error rate $\varepsilon$, the number of replicas grows only logarithmically with the system size, and thus in practice the learning algorithm works with reasonable resources.

Before describing the results of applying 'annealing with learning' to computations with both fixed and mixed boundary conditions, in Fig. 6 we plot the average local overlaps of 2,000 replicas with the solution for a fixed circuit and boundary condition before applying the learning algorithm. The agreement with the data presented in Fig. 4 substantiates the fact that the local majority rule implemented through the independent annealing of the replicas recapitulate the behaviour of the correct solution to the computational problem.

We start from fixed-input boundary condition. Using the algorithm described above, we choose $\tau_a = 2^{13}$ for each iteration, and set the majority rule threshold at $\alpha = 0.7$. In Fig. 7 we show the local order parameter of the final states averaged over 2,000 replicas after each iteration. We emphasize that even though we are plotting the average overlap with the actual solution as a benchmark, in the learning algorithm no reference to $\{q^{\text{sol}}\}$ is made. The weight of each possible state of each gate in the circuit is computed solely from the replicas. After each iteration, with $\tau_a = 2^{13}$ the correlation length grows to $\ell(\tau_a) \sim 10$, and by pinning gates with high percentage of agreement on certain states we are pushing the 'boundary' forward until all gates are fixed. Since the total number of iterations $n_a$ scales linearly with the circuit depth $W$, $n_a \propto W$, the total time to solution $\tau = n_a\tau_a$ also

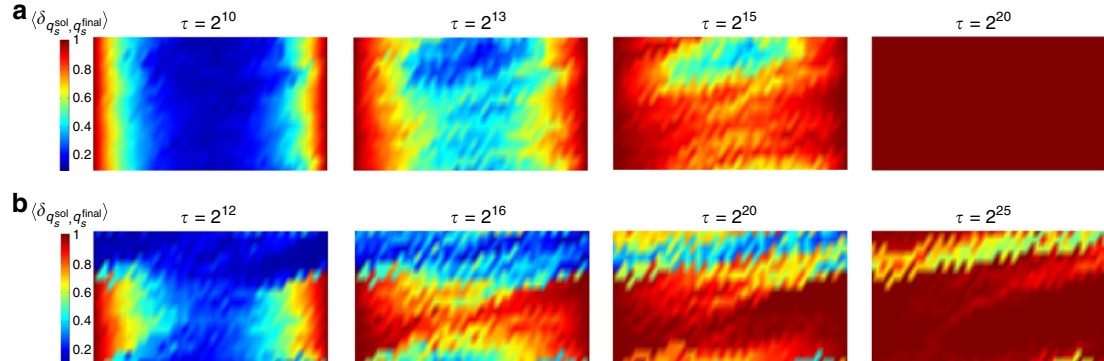

**Figure 6 | Growing correlation length without learning.** The average local overlaps $\langle \delta_{q_s^{\text{sol}}, q_s^{\text{final}}} \rangle$ of 2,000 replicas with $\{q^{\text{sol}}\}$ for a given circuit and boundary state without learning. The system size is $16 \times 42$, with 20% TOFFOLI gates. (**a**) Fixed-input and output and (**b**) mixed boundary conditions.

scales linearly with $W$, $\tau \propto W\tau_a$, consistent with the expectations for the time-to-solution for forward computation. For the computation shown in Fig. 7, it is clear that the 'annealing with learning' process proceeds ballistically and reaches solution with $n_a = 9$ steps.

Now we look at mixed boundary conditions. The results presented in Fig. 8 are obtained by applying the learning algorithm with $\alpha = 0.7$. However, in the case of mixed boundary conditions the process of 'learning' proceeds through two series of annealing steps with different time scales: an initial set of iterations with $\tau_a = 2^{13}$ which build longitudinal correlations required for learning, followed by a set of longer annealing steps with $\tau_b = 2^{18}$ that allow the slower correlations along the transverse direction to develop. Figure 8 shows the progression to solution, which could not be reached for the same computation using the 'vanilla' thermal annealing for our longest accessible times ($\tau \sim 2^{25}$).

We note that this protocol can also be used to solve problems with a 'few', $\mathcal{O}(1)$, solutions. This is best illustrated for the case of two solutions, which can be addressed by carrying out $2n$ computations with mixed boundary conditions, where $n$ is the number of unknown bits in the input. The idea is to define $2n$ problems by fixing each bit at a time to be 0 or 1, while leaving the other $n-1$ bits floating. Since the two solutions must differ in at least one of the $n$ bits, after at most $2n$ steps, this scheme transforms the problem into two separate problems, each of which can be solved by the techniques discussed in this paper. An important problem that falls precisely within this case is factorization of semi-prime numbers $s = p \times q$, where there are exactly two solutions, corresponding to the two ordered pairs $(p,q)$ and $(q,p)$ of primes $p,q$ (assumed to be different).

Finally, we turn to the analysis of cases with multiple solutions and no solution. In both of these cases it is not sensible to compute the local overlap with a solution, as we did for circuit problems with only one solution. Instead, we plot the largest weight of each gate state in the circuit obtained from 2,000 replicas. This is shown in Fig. 9 for an instance with eight solutions obtained by fixing fewer gates (than in the single-solution case) on each boundary; and an instance with no solutions, obtained by fixing a few gates on one boundary to the wrong states. Figure 9 shows that the learning algorithm eventually gets stuck when the replicas cease to agree on gate assignments above the threshold $\alpha$. We note that the learning algorithm cannot differentiate between these two cases. We interpret the freezing of the system as an effect of frustration in satisfying the local gate constraints in the bulk induced by incompatible boundaries in the case of no solution or compatible but competing boundaries in the case of multiple solutions.

**Mapping onto the D-Wave Chimera graph for quantum annealing.** We close this paper by describing a scheme for 'programming' our vertex model into a quantum annealer. In particular, we present an explicit embedding of the tile model of universal classical computing circuits into the Chimera graph architecture of the D-Wave machine. The idea is to use one unit cell to represent one square tile of our construction presented previously. Rectangular tiles (that is, TOFFOLI gates) can be viewed as consisting of two square tiles, thus requiring two unit cells to be embedded in the Chimera graph. We then implement the Hamiltonians of equations (5–7) using the programmable couplers available in the D-Wave machine, as illustrated in Fig. 10 and described in more detail in Methods section.

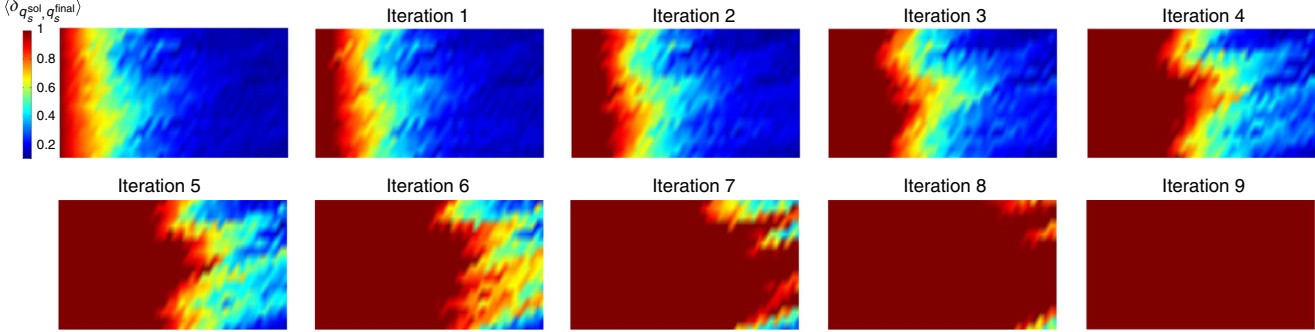

**Figure 7 | Growing correlation length with learning for fixed-input boundary condition.** Annealing with learning for the fixed-input boundary case for a system of size $16 \times 42$ with 20% TOFFOLI gates. The annealing time within each iteration is $\tau_a = 2^{13}$ and the gate state probability threshold $\alpha = 0.7$.

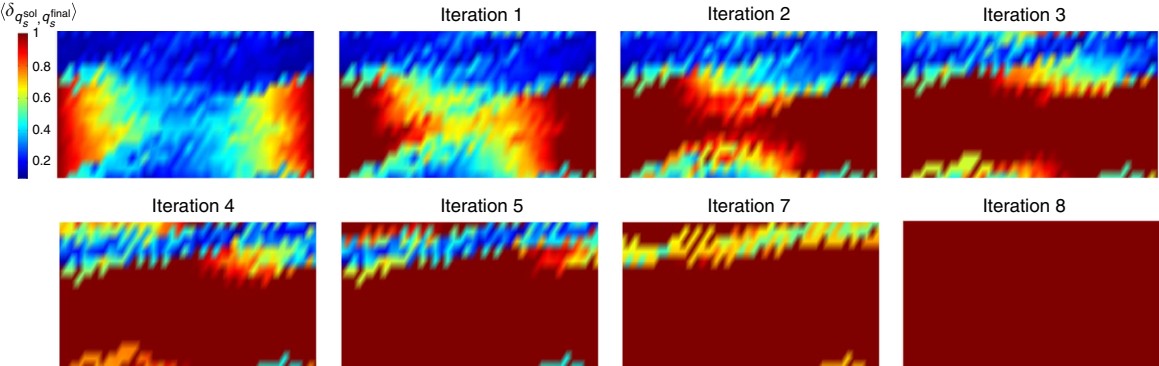

**Figure 8 | Growing correlation length with learning for mixed boundary conditions.** Annealing with learning for mixed boundary conditions and systems of size $16 \times 42$ with 20% TOFFOLI gates. The annealing time within each iteration is $\tau_a = 2^{13}$, and the probability threshold $\alpha = 0.7$. After iteration 6 (not shown), the correlations fully build up along the longitudinal direction, $\tau_a$ is then increased to $\tau_b = 2^{18}$.

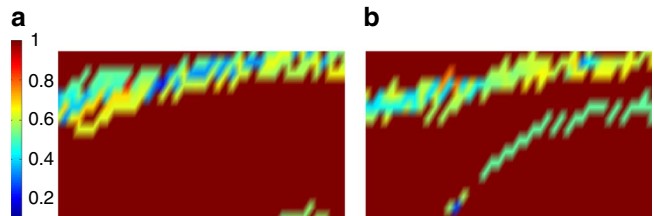

**Figure 9 | Cases with multiple solutions or no solution.** Colour plot of the largest weight of each gate state in the circuit for a system of size 16 × 42 and 20% TOFFFOLI gates after a relaxation time $\tau \approx 2^{20}$. (**a**) Case with eight solutions and (**b**) case with no solution. The learning algorithm eventually gets stuck at the point where no more gates have majority weight above the threshold $\alpha$.

## Discussion

The results of this paper were motivated by an attempt to use our statistical mechanics intuition about lattice models of spin systems to uncover some of the salient features of universal classical reversible computation. There are questions posed and open problems raised by these studies. Here we list four that we find most important.

First, one should understand the scaling of time-to-solution of the various schemes discussed here, including those that utilize learning, as a function of input size and depth for specific computational problems. Under a trivial reduction scheme, one can solve problems with two solutions using similar annealing with learning techniques that we deployed for problems with a unique solution. As an important application we are already investigating the problem of the factorization of semi-primes. The scaling properties of the time-to-solution in the context of this concrete and relevant problem should be contrasted to that obtained in the random circuit with the same concentration of TOFFOLI gates.

A second question raised by our work is the nature of the zero-temperature quantum phase transition encountered in the quantum vertex model, as depicted in Fig. 3. We demonstrated that, in the limit of the trivial computational circuit with no TOFFOLI gates, this transition is second order, in direct analogy to the case of the one-dimensional Ising model in a transverse magnetic field. Whether the transition remains second order or becomes first order for realistic computations (corresponding to a finite concentration of TOFFOLIs) has very important consequences for solutions of computational problems via quantum annealing.

Third, the computational problems discussed here should also be studied directly in a *bona fide* quantum annealer. An important result of this paper is the programming of generic reversible computational circuits into the Chimera architecture of the D-Wave machine. This paves the way for using this type of hardware to study annealing protocols along the $\delta$ axis, as well as arbitrary directions in the $\delta$–$T$ plane. Our approach should also be used as a guide to the development of alternative machine architectures optimized for direct implementations of the vertex model.

Finally, we close with a brief discussion of the broader implications of the mapping of reversible classical computation onto the vertex model on the individual disciplines of computer science and physics. As already mentioned earlier, the line of argumentation in this paper follows a physics perspective, namely, it concentrates on 'typical behaviour' based on heuristic approach to explicit instantiations of the vertex model. Computer science could benefit from further work on more sophisticated theoretical and computational heuristic approaches, special purpose hardware (that is, quantum annealers), and new formal

proofs that rely on statistical mechanics representations of computational problems. At the same time there are lessons to be learned from computer science that we believe may have interesting implications for physics. For example, if $NP \neq P$, the vertex model representing the hardest problems in UNIQUE-SAT can be also viewed as describing a physical glassy system that displays slow dynamics even though the model involves no frustrating interactions, has a unique non-degenerate ground state, and displays no bulk thermodynamic transitions down to zero temperature! There are known examples of systems with glassy dynamics in the absence of a thermodynamic phase transition, such as the kinetically constrained models discussed in refs 30–32. However, the non-Arrhenius relaxation characteristic of these models only translate into a quasi-polynomial time-to-solution of a computational problem. Thus, within the vertex model approach, the existence of hard UNIQUE-SAT problems with exponential or sub-exponential behaviour of the time-to-solution would suggest the existence of a novel family of glassy physical systems without a thermodynamic transition but with exponentially large barriers and corresponding astronomically long relaxation times. This example underscores the richness of the possibilities opened by explorations of the vertex model of classical computation and more generally, of problems at the interface between physics and computer science.

## Methods

**Implementing gates with one- and two-body spin interactions.** We start by representing Boolean variables $x_i = (1 + \sigma_i)/2$ in terms of spins $\sigma_i = \pm 1$ placed on the boundary of each tile, as depicted in Fig. 1. Operations of logic gates are then implemented in a similar way as in ref. 11, by designing a Hamiltonian acting on the spins associated with individual tiles such that (a) the interactions are short ranged and involve at most two bodies and (b) spin (that is, bit) states that satisfy the gate constraint are ground states of the tile Hamiltonian and all other 'unsatisfying' spin states are pushed to high energies.

*Identity (ID) gate.* The ID gate takes two bits $(a, b)$ into $(a, b)$. This is easily enforced by adding ferromagnetic interactions ($J > 0$) that align input bits $a$ and $b$ to output bits $c$ and $d$, respectively, leading to an energy

$$E_{\text{ID}}(\sigma_a, \sigma_b; \sigma_c, \sigma_d) = -J(\sigma_a \sigma_c + \sigma_b \sigma_d). \tag{5}$$

*SWAP gate.* The SWAP gate takes $(a, b)$ into $(b, a)$, and can be implemented in the same manner as the ID gate through a ferromagnetic interaction ($J > 0$),

$$E_{\text{SWAP}}(\sigma_a, \sigma_b; \sigma_c, \sigma_d) = -J(\sigma_a \sigma_d + \sigma_b \sigma_c). \tag{6}$$

*TOFFOLI gate.* The TOFFOLI gate is represented by a rectangular tile with the three input bits $(a,b,c)$ and three output bits $(a', b', d)$ placed on the boundary, as shown in Fig. 1. Notice that in this case we also place an additional ancilla bit in the centre of the rectangular tile, which is essential to satisfy the gate constraint with no more than two-body interactions. The TOFFOLI gate takes the three-bit input state $(a, b, c)$ into $(a, b, ab \oplus c)$. The copying of the first two input bits from the input into the output is accomplished as before through a ferromagnetic coupling: $-J(\sigma_a \sigma_{a'} + \sigma_b \sigma_{b'})$. Enforcing the third output bit $d = ab \oplus c$ requires a more involved interaction. We present the result below, and leave the detailed justification for Supplementary Note 1. The complete energy cost associated to the TOFFOLI gate reads

$$
\begin{aligned}
E_{\text{TOFFOLI}}(\sigma_a, \sigma_b, \sigma_c; \sigma_{a'}, \sigma_{b'}, \sigma_d; \sigma_S) = \\
-J(\sigma_a \sigma_{a'} + \sigma_b \sigma_{b'}) + J(\sigma_a - 3\sigma_b - 2\sigma_c + 2\sigma_d + 4\sigma_S) \\
+ J(-3\sigma_a \sigma_b - 2\sigma_a \sigma_c + 4\sigma_b \sigma_c + 2\sigma_a \sigma_d - 4\sigma_b \sigma_d - 4\sigma_c \sigma_d) \\
+ 4\sigma_a \sigma_S - 8\sigma_b \sigma_S - 6\sigma_c \sigma_S + 6\sigma_d \sigma_S).
\end{aligned}
\tag{7}
$$

**The global constraint and coupling of adjacent tiles.** In addition to satisfying each gate separately, spins shared by neighbouring tiles must be matched across the entire system in order for the tile model to accurately represent the desired computational circuit. To be precise one can imagine splitting each boundary spin into two 'twin' spins and identifying input/output spins with each tile. Within this picture, adjacent spins at the boundary between tiles must be locked together, a constraint we implement by introducing a ferromagnetic 'grout' coupling $K > 0$ between spins on adjacent tiles. The corresponding term in the energy is then written as

$$E_{\text{grout}}(\{\sigma\}) = -K \sum_{\langle i, j \rangle} \sigma_i \sigma_j, \tag{8}$$

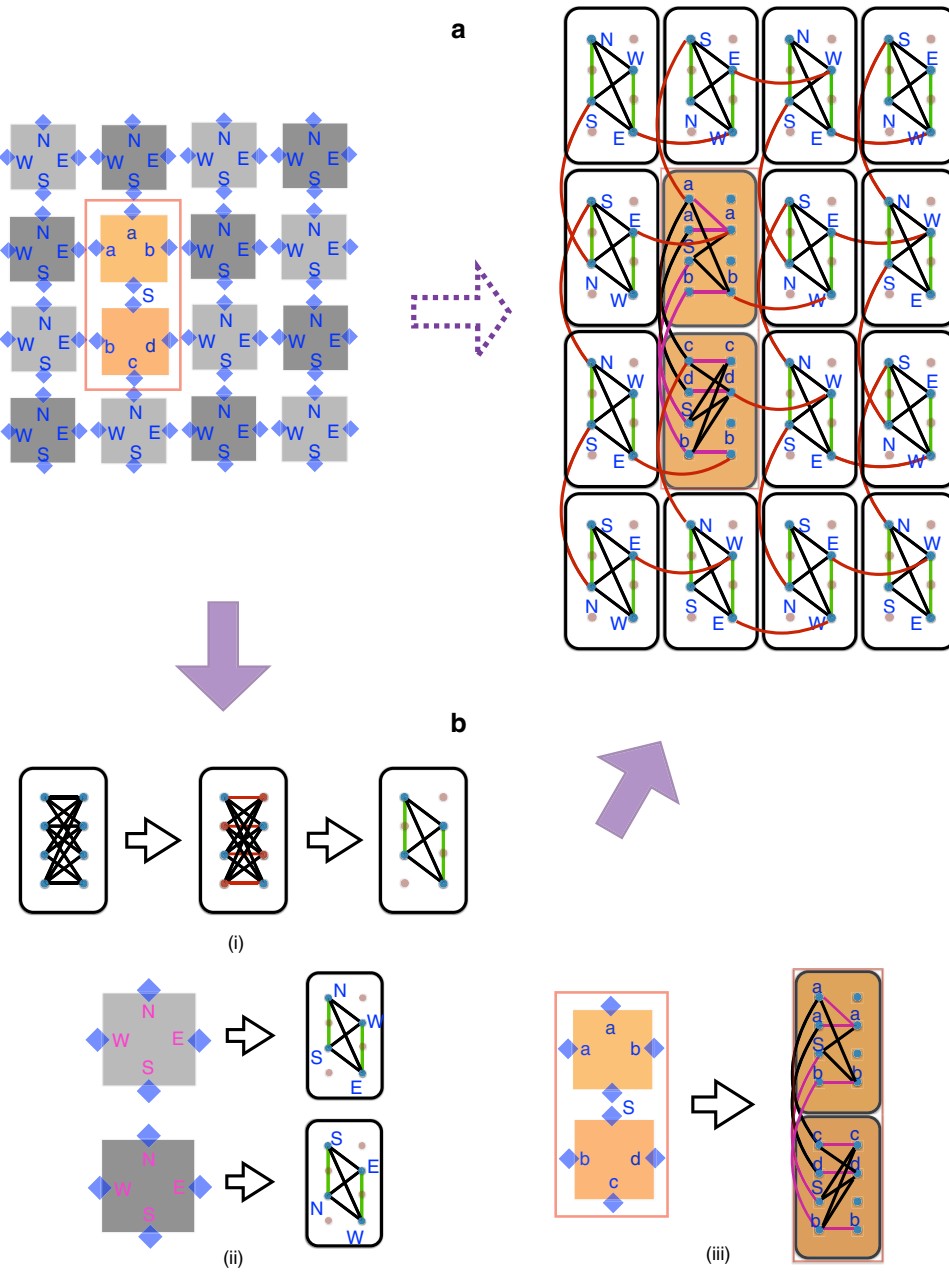

**Figure 10 | Mapping onto the Chimera graph.** Procedure for embedding a 4 × 4 tile lattice into the Chimera graph. (**a**) Left: a generic tile lattice rotated by 45°. Spins are put on the boundary of each tile. The lattice can be further divided into two sublattices, depicted by dark and light grey respectively; right: embedding of the tile lattice into the Chimera graph. The 'grout couplings' are indicated by red links. (**b**) Embedding of each gate into the unit cells of the Chimera graph. (i) Left: a $K_{4,4}$ unit cell of the Chimera graph; middle: to couple qubits in the same column, we slave the qubits to their neighbours in the other column using additional ferromagnetic couplings indicated by red links; right: effectively we are left with four qubits that are fully connected. For simplicity, we hereafter denote the effective couplings between spins in the same column by a single green link. However, one should keep in mind that they are obtained by slaving the spins to the opposite column via large ferromagnetic couplings. (ii) The four qubits in the rotated square tile are labelled by their locations on the tile: N (North), S (South), W (West) and E (East). Tiles corresponding to different sublattices must be embedded differently due to the special connectivity of the Chimera graph. (iii) Embedding of the TOFFOLI gate consisting of two square tiles into two unit cells. (a,b,c,d) corresponds to the input and output bits of the gate, and S is the ancilla bit. In the unit cell, ferromagnetic couplings that copy spins are indicated by purple links, and couplings required in Hamiltonian (7) are indicated by black links.

where $\langle i, j \rangle$ labels pairs of 'twin' spins $i$ and $j$ on the boundary between two adjacent tiles and the sum ranges over all such pairs of the system.

**Boundary conditions.** Completing the description of the 2D model of universal classical computation requires a discussion of boundary conditions, which determine the type of computational problem one is addressing. For example, if the $N$-bit input is fully specified and one is interested in the output, all that is needed is to transfer the information encoded into the input left to right by applying sequentially the gates one column of tiles at a time. In this case, if the depth (that is,

the number of steps) of the computation is a polynomial in $N$, this column–column computation reaches the output boundary, and thus solves the problem, in polynomial time.

As mentioned earlier, by using reversible gates one can also represent computational problems with mixed input–output boundary conditions for which only a fraction of the bits on the left (input) edge and a fraction of the bits on the right (output) edge are fixed. A concrete example is the integer factorization problem implemented in terms of a reversible integer multiplication circuit. A reversible circuit for multiplying two $N$-bit numbers $p$ and $q$ can be constructed

using $5N + 1$ bits in each column. One needs two $N$-bit registers for the two numbers $p$ and $q$ to be multiplied, one $N$-bit carry register $c$ for the ripple-sums, a $2N$-bit register $s$ for storing the answer $p \times q = s$, and one ancilla bit $b$. For multiplication, one only fixes the boundary conditions on the input: $p$ and $q$ are the two numbers to be multiplied, and $c$, $s$ and $b$ are all 0's. For factorization we must impose mixed boundary conditions: on the input side the $c$, $s$ and $b$ registers are fixed to be all 0's; on the output side the $s$ register is now fixed to the number to be factorized, and $c$ and $b$ are again all set to 0. Thus, $3N + 1$ bits in the input and output are fixed, while $2N$ bits are floating on both boundaries.

Boundary conditions on inputs, outputs, or both are imposed by inserting longitudinal fields at the appropriate bit sites, namely,

$$E_{\text{boundary}}(\{\sigma\}) = - \sum_{i \in \text{boundary}} h_i \, \sigma_i, \qquad (9)$$

with $|h_i| = h \gg J$. The sign of an individual $h_i$ field determines the value of the spin $\sigma_i$ and thus of the binary variable $x_i$: For $h_i > 0$, $x_i = 1$, while for $h_i < 0$, $x_i = 0$. If no constraint is imposed on a binary variable $x_i$, then $h_i = 0$.

**Construction of the vertex model.** Combining the contributions above leads us to a classical Hamiltonian that includes the energy functions internal to each tile, the coupling between the spins at the boundary between adjacent tiles, and the magnetic fields associated with the input and output bits defining the boundary conditions of the computation, namely,

$$H_C = \sum_g E_g^J \left( \{\sigma\}_g \right) - K \sum_{\langle i, j \rangle} \sigma_i \, \sigma_j - \sum_{i \in \text{boundary}} h_i \, \sigma_i, \qquad (10)$$

where $\{\sigma\}_g$ labels all the spins and $E_g^J(\{\sigma\}_g)$ represents the energy function of tile (that is, gate) $g$.

This Hamiltonian is the starting point for our mapping of universal classical computation into the Chimera architecture of the D-Wave machine, one of the important results of the paper, which we discuss in detail below. To anticipate the fact that quantum rather than classical thermal annealing may be a more effective way of reaching the ground state and therefore the solution of these computational problems, we add a transverse magnetic field $\Gamma$ to equation (10) to obtain the quantum Hamiltonian

$$\hat{H} = \sum_g E_g^J \left( \{\hat{\sigma}^z\}_g \right) - K \sum_{\langle i, j \rangle} \hat{\sigma}_i^z \, \hat{\sigma}_j^z - \sum_{i \in \text{boundary}} h_i \, \hat{\sigma}_i^z + \Gamma \sum_i \hat{\sigma}_i^x. \qquad (11)$$

However, we find it more expedient and intuitive to work directly with tiles which satisfy the logic gate constraint exactly. We thus proceed by projecting the system onto the manifold of states where all local gate constraints are satisfied by working in the limit in which both $h$ and $J$ are very large; and we imagine varying $K$ and $\Gamma$, with $K, \Gamma \ll J, h$ (ref. 33). This limit is best understood if we switch off the coupling between tiles, $K$. Within a given tile, the configurations that satisfy the logic gate constraints span the degenerate ground-state manifold, while the unsatisfying configurations have energies of order $J$ and higher. Let $\{|q_a\rangle\}$, $a = 1, \ldots, r$ be all the $r$ states spanned by the spin configurations $|\sigma_1, \ldots, \sigma_n\rangle$ that define the ground-state manifold. For two-bit (four-spin) gates we have $r = 4$, while for three-bit (six-spin) gates $r = 8$.

As long as $\Gamma \ll J$ we can understand the effect of a transverse field $\Gamma$ on the $r$ degenerate states by degenerate perturbation theory. Since for reversible gates maintaining the gate constraints requires at least two spin flips, the transverse field $\Gamma$ induces an effective, second-order or higher spin–spin interaction on the ground-state manifold of a given tile of order $\Delta = \Gamma^2/J$ or lower. This discussion leads naturally to the quantum vertex model presented in equation (1).

Switching on the $K$ coupling penalizes configurations in which the states of adjacent tiles are incompatible. Thus, to satisfy both intra- and inter-tile constrains that define the computational process we must reach the limit of $\Delta \ll K \ll J, h$.

**Thermodynamics of the classical vertex model.** We start by considering the partition function of the classical limit of the Hamiltonian in equation (1), that is, $\Delta = 0$, which we obtain via a transfer matrix calculation. Consider first a system with free boundary conditions at both ends. The partition function for the vertex model can be more easily written using the spin variables on the links of the lattice. Let $\{\sigma\}_j$ denote the spin states on a vertical line, which cuts across $3L$ spins (with $L$ the number of vertices or 3-bit gates in a column). For convenience we shall utilize the notation $|\{\sigma\}_j\rangle$ for the vectors in this transfer matrix calculation. Within this notation, one can write the partition function as

$$Z = \sum_{\{\sigma_1\}, \ldots, \{\sigma_{2W}\}} \langle \{\sigma_1\} | P_1 | \{\sigma_2\} \rangle \langle \{\sigma_2\} \} | T | \{\sigma_3\} \rangle \times \cdots \times \langle \{\sigma_{2j-1}\} | P_j | \{\sigma_{2j}\} \rangle$$
$$\langle \{\sigma_{2j}\} | T | \{\sigma_{2j+1}\} \rangle \times \cdots \cdots \times \langle \{\sigma_{2W-1}\} | P_W | \{\sigma_{2W}\} \rangle, \qquad (12)$$

where the matrix $T$ encodes the energy costs for matching spins across the links, and the matrices $P_j$ encode the computations performed by one column of gates. The two types of slices are depicted in Fig. 2. Notice that the $T$ is the same for all

slices, and its matrix elements are given by

$$\langle \{\sigma_{2j}\} | T | \{\sigma_{2j+1}\} \rangle = \exp\left( \sum_{a=1}^{3L} \beta K \sigma_{2j,a} \sigma_{2j+1,a} \right), \qquad (13)$$

whereas $\langle \{\sigma_{2j-1}\} | P_j | \{\sigma_{2j}\} \rangle$ represents the matrix element of $P_j$ at the $j$th column and thus depends on the particular set of gates within that column. However, all $P_j$ are permutation matrices since all gates are reversible. This fact is essential because it allows us to compute the partition function exactly, irrespective of the circuit.

For the next step notice that the vector $|\Sigma\rangle = \sum_{\{\sigma\}} |\{\sigma\}\rangle$ is an eigenvector of $P_j$ for any operation $P_j$:

$$P_j |\Sigma\rangle = P_j \sum_{\{\sigma\}} |\{\sigma\}\rangle = \sum_{\{\sigma\}} P_j |\{\sigma\}\rangle = \sum_{\{\sigma'\}} |\{\sigma'\}\rangle$$
$$= |\Sigma\rangle, \qquad (14)$$

where we used that we can relabel the states after the permutation. The vector $|\Sigma\rangle = \sum_{\{\sigma\}} |\{\sigma\}\rangle$ is also an eigenvector of $T$:

$$T |\Sigma\rangle = T \sum_{\{\sigma\}} |\{\sigma\}\rangle = \sum_{\{\sigma\}, \{\sigma'\}} |\{\sigma'\}\rangle \langle \{\sigma'\} | T | \{\sigma\} \rangle$$
$$= \sum_{\{\sigma'\}} |\{\sigma'\}\rangle \sum_\sigma \exp\left( \sum_{a=1}^{3L} \beta K \sigma_a' \sigma_a \right)$$
$$= \sum_{\{\sigma'\}} |\{\sigma'\}\rangle \prod_{a=1}^{3L} \sum_{\sigma_a = \pm 1} e^{\beta K \sigma_a' \sigma_a}$$
$$= \sum_{\{\sigma'\}} |\{\sigma'\}\rangle \, (2 \cosh \beta K)^{3L}$$
$$= (2 \cosh \beta K)^{3L} |\Sigma\rangle. \qquad (15)$$

By collecting all the factors we arrive at the partition function

$$Z = \langle \Sigma | P_1 \, T \, P_2 \, T \, \ldots \, T \, P_W | \Sigma \rangle$$
$$= (2 \cosh \beta K)^{3L(W-1)} \langle \Sigma | \Sigma \rangle. \qquad (16)$$

The overlap $\langle \Sigma | \Sigma \rangle = 2^{3L}$ reflects the $2^{3L}$ degenerate ground states corresponding to open boundary conditions on both boundaries. Had we fixed one of the boundaries to a particular state $|\{\sigma\}_{\text{fixed}}\rangle$ we would have instead obtained an overlap $\langle \{\sigma\}_{\text{fixed}} | \Sigma \rangle = 1$. More generally, in the thermodynamic limit boundaries contribute an entropic term that counts the number of ground states, but does not affect the bulk thermodynamics. In particular, the bulk free energy is that of a paramagnet:

$$\beta F = - [3L(W-1)] \ln(2 \cosh \beta K). \qquad (17)$$

This also implies that thermodynamics alone, which is independent of the specific form of the circuit, cannot reveal the complexity of a ground-state computation, which is reflected in the dynamics of the system's relaxation into its ground state.

**Metropolis algorithm for thermal annealing.** The Metropolis simulations are carried out as follows. We work on a lattice of $L \times W$ vertices, using the Hamiltonian of equation (1) with $\Delta_{q_s, q_{s'}} = 0$. Periodic boundary conditions are used in the transverse direction, that is, the circuit is laid down on the surface of a tube of length $W$ and circumference $L$. We consider four types of circuits corresponding to different concentrations of TOFFOLI gates (the other four types of gates are assigned equal concentrations): a circuit with only TOFFOLI gates (100% concentration), and random circuits with 40, 20 and 0% concentration of TOFFOLI gates.

The first step of the simulation is to construct a reference state $\{q^{\text{sol}}\}$ that solves the circuit, by fixing the states $q_s^{\text{sol}}$ for the vertices $s$ at the left boundary, and determining and storing all other states $q_s^{\text{sol}}$ for vertices $s$ in the rest of the circuit. Next, we construct three explicit boundary conditions consistent with the reference state, $\{q^{\text{sol}}\}$, that will serve as the input states for our simulations: (1) fixed input, for which we apply pinning fields at the left boundary that fix the states to match $q_s^{\text{sol}}$ for the vertices $s$ at the left boundary, and leave the other boundary free (no pinning field); (2) fixed input and output, for which we pin all vertices $s$ on the left and right boundaries to those defined by $q_s^{\text{sol}}$; and 3. mixed boundaries, for which we pin $L_\partial = L/2 + 3$ vertices on both the left and right boundaries to the solution values $q_s^{\text{sol}}$, but leave all the remaining boundary vertices free. Our computations proceed in each of these three cases by averaging over 2,000 independent random instances of input states for a given circuit with a fixed concentration of TOFFOLI gates corresponding to different $\{q^{\text{sol}}\}$ and computing the average order parameter $\langle m \rangle$ as a function of the relaxation time $\tau$. Here it is important to stress that the partial specification of boundaries in the case of mixed boundary conditions generically leads to multiple solutions which compete in establishing the local configurations of gates consistent with the global constraints defining the computational circuit. While we also discuss cases with multiple solutions and no solution in Results, the focus of this paper is on problems with a single solution. To ensure a single solution in the case of mixed boundary conditions we always check that each of the random instances of the input state for a given circuit allows for one and only one solution.

**Mapping onto the Chimera architecture of the D-Wave machine.** Figure 10 shows the 'flow chart' of embedding a $4 \times 4$ tile lattice into the Chimera graph. The entire tile lattice is rotated by 45° for convenience, and spins living on the boundary of each tile are shown explicitly. The lattice of tiles can be further divided into two sublattices labelled by dark and light grey, for reasons that should become clear shortly. Now let us first consider how to encode the ID and SWAP gates represented by a single square tile into a unit cell. The embedding involves internal couplings $J$ that enforce the gate constraints, and the 'grout' couplings $K$ that match adjacent tiles. The Chimera unit cell forms a complete bipartite graph $K_{4,4}$, as depicted in Fig. 10b-(i), with each spin in one column coupled to all spins in the other, but not to those in their own column[34]. To obtain the generic spin couplings to represent the gates on a single tile, which inevitably involves couplings between qubits in the same column as well, we use an additional ferromagnetic coupling to slave the spins in one column to their nearest neighbours in the other column. Thus, effectively we are left with four spins that are fully connected. Details are shown in Fig. 10b-(i).

To use the connectivity of the Chimera graph architecture and couple adjacent tiles properly, it is convenient to explore the bipartiteness of the square lattice. Let us take one tile from the rotated tile lattice, and label the four qubits by their locations on the tile: N (North), S (South), W (West) and E (East), as shown in Fig. 10b-(ii). The spins in the adjacent tiles are matched by the 'grout' coupling $K$. Upon a careful inspection of the resulting tile lattice, we notice that the qubits labelled by N and W in one tile are always connected respectively to qubits S and E in its neighbour. Therefore, once we fix the embedding of one sublattice in the unit cell, the embedding of the other sublattice must be different, because qubits in one unit cell are only coupled to those at the same place in the neighbouring unit cell. We map the two sublattices of the tile lattice in the unit cells as illustrated in Fig. 10b-(ii).

Finally, let us show how to embed the TOFFOLI gate, which corresponds to a rectangular tile, into the unit cells. The TOFFOLI gate can be viewed as consisting of two square tiles, thus requiring two unit cells, and the spins coupled between these two tiles exactly provide the ancilla bit needed in Hamiltonian of equation (7). Similar to the square tiles considered above, within a unit cell we use additional ferromagnetic couplings to slave qubits from one column to the other when necessary. The explicit mapping is shown in Fig. 10b-(iii). As we have already seen, to come up with a proper embedding, one has to carefully take into account how qubits are coupled to adjacent unit cells. Putting all of the above ingredients together, we arrive at the embedding of the entire $4 \times 4$ tile lattice including TOFFOLI into the Chimera graph, as shown in Fig. 10a.

**Data availability.** The data that support the main findings of this study are available from the corresponding author upon request.

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

## Author contributions

All authors contributed to all aspects of this work.
