## [Peer Review File · Nature Communications]

Reviewer #1 (Remarks to the Author):

This manuscript presents a way to map reversible classical computations to a vertex model, whose size is the width times the depth of the original calculation. The proposal is to solve this vertex model and find its ground state - and thus perform the encoded computation. While theoretically interesting as yet another way of mapping computational problems to statistical physics models, I do not see any important results.

First, it has to be noted that the forward calculation of the reversible circuit can be done with a (space-time) effort linear in the size of the vertex model. Since this is also the minimal effort needed to solve the model in the best case there is no advantage in this case.

The authors point out that the same approach can also solve the inverse problem, where the output is given. This is, in principle, more interesting. However now it becomes a question of what is the complexity of solving this problem. While the authors argue that the model has no phase transition, the dynamics can still be exponentially (or double exponentially) slow. The numerical evidence provided is for problems far too small to reach the scaling regime, and even then the results are inconclusive.

In the last section the authors argue that "quantum annealing is a powerful approach that, in principle, should speed up computations by orders of magnitude". This is, as the authors later write "wishful thinking". There is to date no evidence that quantum annealers scale better than classical algorithms for solving computational problems. Furthermore, if the "orders of magnitude" refer to the cited recent paper by the Google group, then the authors should keep in mind that the same paper states that classical algorithms exist which solve the same problems even faster.

The authors then argue that due to power law behavior at a quantum phase transition, quantum annealing should be able to solve these problems in polynomial time. This completely ignores the well-known fact (see e.g. S, Knish, arXiv:1506.08608) that the computational bottlenecks are typically not at the initial second order quantum phase transition, but at avoided level crossing with exponentially small gaps after the quantum phase transition. Hence all these arguments remain just "wishful thinking" as the authors mention themselves.

Hence, while the original mapping may be interesting to a small group of scientists working on mapping of computational problems to statistical models, I do not see much relevant results for a high-impact journal.

Reviewer #2 (Remarks to the Author):

In "Solving Classical Computational Problems by Annealing a Planar Quantum Vertex Model", authors Chamon, Mucciolo, Ruckenstein, and Yang present the results of the study of a class of statistical physics models in which the ground state of the system encodes a computation. The paper is mainly up of four major parts, in the first the authors define a two-local spin system whose ground state encodes a reversible computation, in the second part the author presents a "vertex model" version of this first model and computes exactly its partition function, in the third the authors perform numerical simulations of this model in order to establish the computational complexity of the model, and in the final part the authors embed these models on a graph relevant for quantum adiabatic algorithms. The positives of this paper are that the authors define an interesting model, the quantum vertex model, whose ground state encodes a classical computation, and which is amenable to exact solution. For this model the fact that it has no finite temperature phase transition and that how one gets into this thermal state is related to the computational complexity is a nice result. However the paper also has some serious problems that I feel the authors need to address. These are threefold:

1) The paper does not do an adequate job delineating which of the results in the paper concern the (more physically realistic) two spin models in section II and section V and the vertex model defined in section III and section IV. In particular it is not at all clear that the vertex model is thermodynamically connected to these other models. The authors argument that a transverse field acting on the two spin models will act in this manifold is correct, however it not at all clear why the vertex model thermodynamics and dynamics should be related. In reading the introduction and conclusion the authors make little distinction between these two models.

2) The paper uses very bad proxies for difficult of problems when analyzing the different regimes of the model and this renders their conclusions about computational complexity not very useful. In section IV the authors consider three different classes of problems, those with all Tofolli gates, those with 20% random gates, and those with 20% random gates, but no Tofolli gates (and so not universal). The main motivation for this is to study mixed boundary cases where these problems are related to problems who computational complexity is NP-hard. This however, is quite adequate. This is because computational complexity is defined for an entire class of problems, and it is not clear that the problems that these random instances come from are themselves NP-hard. In particular it is well known that there in randomized instances of SAT problems there are regimes where these problems are easy and those where it is hard. The authors need to use this literature to justify why the problems they are examining are in the hard regime.

3) On page 5 the authors make a claim that the fixed boundary method can be used to factor. However I do not believe their method works. The basic idea the authors have is that if takes a circuit that multiplies two numbers and fixes the output spins to be the number to be factored, then one can read off the factorized numbers from the input. The authors circuit for this problem takes two number p and q and produces a reversible circuit that outputs the product $N=pq$ as well as garbage bits for the ripple sums and an extra ancilla bit. The authors claim that setting these ancilla and carry sum bits to zero in both initialization and the output will result in a ground state which factors. However, this is not at all clear because, for a given p and q , the ripple carry bits may not be zero. The authors should justify why setting these bits to zero leads to the factored number.

Another note is that while the authors are clearly motivated by using their setup to solve hard computational problems in polynomial time, they don't have any real evidence that their system can do this. The scaling they obtain is completely inconclusive, and their is no intuitive or logical reason given for why this system should be able to solve these problems.

Finally here are some issues that make the paper difficult to read or are unclear.

1) The authors two-local system whose ground state implements the reversible Tofolli gate uses an ancilla bit. This bit's state is different for different inputs and outputs of the ground state. It is not clear how to interpret this. If the ancilla is included in the input our output, then the circuit is not reversible.

2) The authors should reference and compare their results to those in "Ground State Spin Logic" EPL 99 (2012) 57004 by J. D. Whitfield, M. Faccin, J. D. Biamonte.

3) p1. "As we argue in more detail below, this should allow us to tackle a subclass of the Circuit Satisfiability (CSAT) problem and to solve factoring problems by using multiplication circuits with polynomial depth". The word "tackle" is very vague, the authors should clearly state that their approach allows one to formulate these problems in terms of the behavior of their physical system, not that they can "tackle" the problem.

4) p1. "Here we focus on reversible rather than irreversible computation in order to address problems with both fixed-input and mixed-boundary conditions on inputs and outputs, as explained above." The paper reference appears to have a finite temperature phase transition (when used with fault tolerant circuits). The authors should comment that while these systems both encode computation in their ground state, they appear to be different from a statistic physics

perspective. (Note also that Ref 17 fixes output bits in ancilla input bits in their complexity theory results, so it is not clear that mixed boundary fixing is novel).

5) p1, "intuitive ideas from statistical mechanics" This should probably be more specific: ideas whose intuition comes from the statistics of spatially local systems.

6) p2. " In particular, the resulting thermodynamic behavior is that of a paramagnet" This contrasts with Ref. 17 and this should be pointed out.

7) p5, 'We present the result below, and leave the detailed justification for the Appendix.' this justification should be made inline here in a condensed form as it is a key feature of the rest of the paper to understand the model.

8) p5, "the calculation can be done straightforwardly in polynomial time via the column-by-column transfer" Please define transfer.

9) p7, "Any circuit can be laid down using these four rectangular tiles plus the TOFFOLI tile" Please use more precise terms than 'laid down'.

10) p8. "Therefore, there are no phase transitions and the thermodynamics alone cannot reveal the complexity of the ground-state calculation, as it does not distinguish trivial problems (e.g., when all gates are ID or SWAP gates)

from complicated ones (e.g., when the circuit contains a finite concentration of TOFFOLI gates)." It would seem that the more important argument is that this doesn't depend on the boundary conditions, since even TOFFOLI gates in the forward direction can do what we consider traditionally not complex computations.

11) p11, m was already used, please don't reuse variable names.

12) p15."The addition of TOFFOLI gates complicates the analysis, but we find any scenario in which the phase transition disappears unreasonable" This seems completely unsubstantiated, one should not interject personal opinion unless clearly delineating that you are speculating.

Reviewer #3 (Remarks to the Author):

The paper "Solving classical computational problems by annealing a planar quantum vertex model" by Chamon et al. is a nice piece of work which provides a very interesting statistical mechanical view of quantum annealing. The introduction of an effective quantum vertex model allows for some detailed analysis of simulated annealing models. Given that their results are novel, I would only ask the authors to couch their results more deeply into the literature and clarify crucial points that were difficult to follow in the manuscript. In what follows, I will point out two places that can be clarified, give suggested references that may further inspire the authors, and finally give miscellaneous improvements.

First, the paper takes a very nice approach inspired by a rich understanding of the vertex model. Hence, equation (8) is critical to understanding the key results of the paper. Thus, it is prudent to elaborate on equation (8) further. The symbols introduced in (8) are not well defined and some of them are unused in the remainder of the paper. Specifically, the term $K_{\{q_s, q_{s'}\}}^{\{g_s, g_{s'}\}}$ is very confusing since there is already a variable K labelling the "grout" coupling. Also in (8) you have $\Delta_{\{q_s, q_{s'}\}}$ which you say for simplicity can be set uniformly but never have an example where the subscripts are used. A graphical representation e.g. over Figure 8, would be most helpful.

Second, the authors try to relay the implications that their work has in the context of computer science complexity theory. Let me note that among NP problems not in P there are few are not proven to be NP-complete. These so called NP-intermediate problems include factoring and graph isomorphism. On page 2 in the sentence "When viewed from a computer science" as well as later in the paragraph containing eq (27) on pg 9, the authors compare their problems to graph isomorphism. While the authors have not made any undue claims, I urge the authors to take even more caution given that their paper didn't draw a clear line between the NP-complete and

suspected NP-intermediate problems. Also on page 15 in the conclusion, I would suggest weakening their claim that "problems expected to be NP-complete may be solvable in quasi-polynomial time."

Concerning missing references, I think the paper has missed at least three key references. Most importantly,

>

> Nonperturbative k-body to two-body commuting conversion Hamiltonians
> and embedding problem instances into Ising spins by Biamonte. Physical
> Review A 77, 052331 (2008)

>

is an earlier than the Crosson paper [17] and is aimed at the task of ground state encodings of classical problems. There was also a follow up by Biamonte and co-authors,

>

> Ground-state spin logic by Whitfield, Faccin, Biamonte. Europhysics
> Letters, vol 99, pg 57004 (2012)

>

that is also relevant to the present article as it also gives constructions similar to those found in section II.A. Finally, there is an early reference on factoring as an optimization problem that contains a more detailed discussion that found on page 5 of your manuscript:

>

> Factoring as optimization by Burges. Microsoft Research
> MSR-TR-2002-83, Technical Report (2002)
> <https://www.microsoft.com/en-us/research/wp-content/uploads/2016/02/tr-2002-83.pdf>

>

Miscellaneous comments:

1. On page, 1st paragraph "implies means" -> "implies"
2. In equations (21) it seems as though 'e' is missing if I use eq (19). This means (22) and (23) are also mistyped.

REPLY TO REFEREE 1 – NCOMMS-16-08852

The referee’s comments made us realize that some of our points were not sufficiently clear, and helped us improve the manuscript.

1. *This manuscript presents a way to map reversible classical computations to a vertex model, whose size is the width times the depth of the original calculation. The proposal is to solve this vertex model and find its ground state - and thus perform the encoded computation. While theoretically interesting as yet another way of mapping computational problems to statistical physics models, I do not see any important results.*

The referee asserts: **“While theoretically interesting as yet another way of mapping computational problems to statistical physics models, I do not see any important results.”** This statement does not acknowledge the limitations of the mappings onto statistical mechanics models that exist in the current literature. Traditional mappings result in random graphs displaying a glass transition that impedes the path to solution. In our mapping, the lack of a thermodynamic phase transition, which we prove explicitly, and the focus on reversible computation, are two important aspects that define a novel statistical mechanics route to classical computation. To clarify this point, in the fourth paragraph of the introduction of the new manuscript we mention the XORSAT, (a problem in P) which maps onto a diluted p-spin model that displays a glass transition. Another important result is the mapping of our model onto the Chimera architecture of the D-wave Machine, which establishes a way of programming quantum annealers to implement our method. Here it is worth noting that, so far, encodings implemented in the D-wave Machine for solving optimization problems are known to lead to glassy phase transitions, a pitfall our mapping avoids.

2. *First, it has to be noted that the forward calculation of the reversible circuit can be done with a (space-time) effort linear in the size of the vertex model. Since this is also the minimal effort needed to solve the model in the best case there is no advantage in this case.*

It seems to us that the referee might have misinterpreted the discussion of forward computation in the original manuscript. While it is obvious that forward computations can be solved in a time linear in the circuit depth we have been stressing that plain annealing is *not* an efficient way to solve the forward computation problem. Our focus was on exploring for the first time how *the simplest annealing* algorithm performed in the vertex model in general, both in the trivial case of forward computation and in more complex problems, with mixed boundary conditions.

Partly in response to the referee’s comment, in the current version we show how annealing can be modified to solve forward problems with an effort linear in the depth of the circuit. We also apply this new type of annealing protocol to speedup solution of more complex problems with mixed boundary conditions, where forward computation with polynomial effort is not an option (for the lack of a polynomial algorithm) but a polynomial verification circuit exists.

3. *The authors point out that the same approach can also solve the inverse problem, where the output is given. This is, in principle, more interesting. However now it becomes a question of what is the complexity of solving this problem. While the authors argue that the model has no phase transition, the dynamics can still be exponentially (or double exponentially) slow. The numerical evidence provided is for problems far too small to reach the scaling regime, and even then the results are inconclusive.*

The comment that the numerical results are not conclusive with respect to the time-to-solution is, of course, correct and already acknowledged in the manuscript. In the current version, we have removed any reference to the scaling of time-to-solution. However, it seems that the referee did not appreciate the fact that *single parameter* scaling is established in our results, even for the relatively small problems we were able to probe. The *collapse* of the data onto universal curves is *nontrivial*, even if we cannot extract the complexity of computation because we cannot currently probe sufficiently long correlation lengths.

4. *In the last section the authors argue that “quantum annealing is a powerful approach that, in principle, should speed up computations by orders of magnitude”. This is, as the authors later write “wishful thinking”. There is to date no evidence that quantum annealers scale better than classical algorithms for solving computational problems. Furthermore, if the “orders of magnitude” refer to the cited recent paper by the Google group, then*

the authors should keep in mind that the same paper states that classical algorithms exist which solve the same problems even faster.

We have a different point of view on the potential importance of quantum annealers, an area still under development. Indeed, as the referee points out the Google paper acknowledges that there are heuristic classical approaches that provide faster times-to-solution than those obtained by using the D-wave machine. The importance of the Google paper is that it shows a substantial speedup over *classical annealing* resulting from an encoding that explores intelligently quantum tunneling. Even if heuristic classical algorithms maybe faster, the direct (“apples-to-apples”) comparison between classical and quantum annealing is important and could also point to ways of speeding up more complex heuristic classical algorithms. Encodings such as ours, that avoid thermodynamic phase transitions (and the pitfalls of glassy phases), may make existing quantum annealers (e.g., the D-wave machine) more efficient, and may also lead to more powerful machine architectures. More generally, our view is that, while not yet fully proven, the exploration of quantum annealers is an important and promising area of future research at the interface between classical and quantum computing. Our paper points to a new, unexplored direction in this rapidly developing field.

5. *The authors then argue that due to power law behavior at a quantum phase transition, quantum annealing should be able to solve these problems in polynomial time. This completely ignores the well-known fact (see e.g. S, Knish, arXiv:1506.08608) that the computational bottlenecks are typically not at the initial second order quantum phase transition, but at avoided level crossing with exponentially small gaps after the quantum phase transition. Hence all these arguments remain just “wishful thinking” as the authors mention themselves.*

The computational bottlenecks discussed in the paper by Knish (arXiv:1506.08608) are not relevant to the discussion of our paper: (a) Knish studies a non-local spin model, whereas our model is local; and (b) Knish’s model displays a second-order transition into a spin glass phase, which is absent in our model. The referee’s comment further underscores the novelty and potential benefit of our approach.

REPLY TO REFEREE 2 – NCOMMS-16-08852

In “Solving Classical Computational Problems by Annealing a Planar Quantum Vertex Model”, authors Chamon, Mucciold, Ruckenstein, and Yang present the results of the study of a class of statistical physics models in which the ground state of the system encodes a computation. The paper is main up of four major parts, in the first the authors define a two-local spin systems whose ground state encodes a reversible computation, in the second part the author present a “vertex model” version of this first model and compute exactly its partition function, in the third the authors perform numerical simulations of thi model in order to establish the computational complexity of the model, and in the final part the authors embed these models on a graph relevant for quantum adiabatic algorithms. The positives of this paper are that the authors define an interesting model, the quantum vertex model, whose ground state encodes a classical computation, and which is amenable to exact solution. For this model the fact that it has no finite temperature phase transition and that how one gets into this thermal state is related to the computational complexity is a nice result. However the paper also has some serious problems that I feel the authors need to be address.

We thank the referee for his/her questions and suggestions, and we are pleased that he/she recognized the novelty of the two key aspects of our approach, namely, the absence of a thermodynamic phase transition and the encoding of the complexity of computations in the dynamics of the relaxation into the ground state. We also appreciate the referee’s comments, which helped us revised and improve the paper. Below we address each of the referee’s comments:

1. *The paper does not do an adequate job delineating which of the results in the paper concern the (more physically realistic) two spin models in section II and section V and the vertex model defined in section III and section IV. In particular it is not at all clear that the vertex model is thermodynamically connected to these other models. The authors argument that a transverse field acting on the two spin models will act in this manifold is correct, however it not at all clear why the vertex model thermodynamics and dynamics should be related. In reading the introduction and conclusion the authors make little distinction between these two models.*

To clarify the discussion of the mapping we placed it all in one section (Section II). This includes (a) the representations of gates as square or rectangular tiles for which corresponding truth tables are encoded in the

ground state of four- and six-spin models, see Eqs. (1)-(3); (b) the coupling of individual tiles (gates) using the “grout” interaction that ensures that the output of one tile agrees with the input to adjacent tiles, see Eq. (4); and (c) the insertion of boundary fields to control input and output bits, see Eq. (5). The resulting energy function (Hamiltonian), which describes both the individual gates and their interaction, defines the “Tile Model”, a statistical mechanics model the ground state of which satisfies all the constraints of the computational circuit, see Eq. (6) for the classical version, and Eq. (7) for the quantum version. It is this “Tile” model that we use to program the Chimera graph of the D-wave machine. In principle, this model explicitly includes excited states, which violate the gate constraints, but these can be eliminated at low temperatures in the Chimera graph by tuning parameters to increase the energy of excited states.

The more efficient way of representing arbitrary computations is to take a limit in which excited states of the “Tile Model” Hamiltonian are essentially projected out and one works in a space in which (a) local gate constraints are satisfied exactly; and (b) the global constraints are enforced by the “grout” Hamiltonian. This limit of the “Tile Model” can be derived exactly and is written in terms of operators, which only act on fully satisfied gates. It is this special limit of the “Tile Model” that we refer to as the “Vertex Model.” They share the same ground state and, in the large- J limit, also have the same low-temperature thermodynamics and dynamics. We note that the virtue of the vertex model is that its dynamics and thermodynamics do not violate the local gate constraints, where as this is only true at low energies in the “Tile Model”. For this reason, most of the manuscript, with the exception of the mapping onto the Chimera graph of the D-wave Machine, uses the vertex model, namely, the “strong J -coupling” limit of the “Tile Model”, to discuss and implement all actual computations.

We hope that, with the reorganization and rewriting of Sec. II, these facts became clearer.

2. *The paper uses very bad proxies for difficult of problems when analyzing the different regimes of the model and this renders their conclusions about computational complexity not very useful. In section IV the authors consider three different classes of problems, those with all Tofolli gates, those with 20% random gates, and those with 20% random gates, but no Tofolli gates (and so not universal). The main motivation for this is to study mixed boundary cases where these problems are related to problems who computational complexity is NP-hard. This however, is quite adequate. This is because computational complexity is defined for an entire class of problems, and it is not clear that the problems that these random instances come from are themselves NP-hard. In particular it is well known that there in randomized instances of SAT problems there are regimes where these problems are easy and those where it is hard. The authors need to use this literature to justify why the problems they are examining are in the hard regime.*

We agree with the referee’s comments on the issue of complexity, and we have made an effort to clarify the issue in the revised manuscript. Indeed, we cannot argue whether or not the random instances of the problem are representative of the hardest problems in the class. What we believe is that we can define them as “typical” computational problems drawn at random from a statistical ensemble of circuits, a common view of physics approaches to be contrasted with the “worst case” scenarios usually discussed in computer science. We note that the well-known discussions of K-SAT problems using belief and survey propagations are very much in the same vein as the approaches adopted in our manuscript. In the revised manuscript, we emphasize the distinction between worst case and typical instances.

Motivated by the referee’s comment, we also tried to frame the discussion on why we could reach solution in problems with a single solution, while our methods did not converge in problems with multiple solutions. (From a physics perspective, the problems with a unique solution correspond to non-competing ground states.) We believe this finding is again related to typicality, because solving the worst-case problems with a single solution is as hard as UNIQUE-SAT, which is proven to be as hard as SAT. Gaining a better understanding as to why the problems with unique ground state appear simpler to our methods, and clarifying the role of typical vs. worst-case scenarios in such problems, is an interesting problem; our work is just a first step in this direction.

3. *On page 5 the authors make a claim that the fixed boundary method can be used to factor. However I do not believe their method works. The basic idea the authors have is that if takes a circuit that multiplies two numbers and fixes the output spins to be the number to be factored, then one can read off the factorized numbers from the input. The authors circuit for this problem takes two number p and q and produces a reversible circuit that outputs the product $N=pq$ as well as garbage bits for the ripple sums and an extra ancilla bit. The authors claim that setting these ancilla and carry sum bits to zero in both initialization and the output will result in a*

ground state which factors. However, this is not at all clear because, for a given p and q , the ripple carry bits may not be zero. The authors should justify why setting these bits to zero leads to the factored number.

The referee's worry that there could be a problem with the carry bits not being zero after the calculation is justified: that these bits *do* return to zero after each stage of the reversible adder pieces is a *nontrivial result of the appropriate design*. If this were not the case, one would need to bring in fresh carry bits and each stage of an adder, and instead of $\mathcal{O}(N)$ bits one would need $\mathcal{O}(N^2)$ bits in the circuit. This problem is circumvented by a design that brings the carry bits (reversibly) back to zero after they are used in a ripple-adder. The circuit is a variant of that of Vedral, Barenco, and Ekert (Ref. 30 of our manuscript), and we have set it up explicitly for multiplication and checked that it works correctly using a $5N + 1$ -bit circuit.

Another note is that while the authors are clearly motivated by using their setup to solve hard computational problems in polynomial time, they don't have any real evidence that their system can do this. The scaling they obtain is completely inconclusive, and there is no intuitive or logical reason given for why this system should be able to solve these problems.

On the other note by the referee: while the *collapse* of the data onto universal curves is *nontrivial*, we agree that our system sizes are not sufficient to conclusively extract time-to-solution as function of system size. In the revised manuscript, using "annealing with learning", we succeeded in bringing systems with mixed boundary conditions all the way to solution. But to get the proper scaling of time-to-solution with system size would require large-scale simulations, in a scale beyond our present computational resources. We thus removed from the manuscript any unsubstantiated reference to the scaling of time-to-solution.

We have also implemented changes that reflect each of the referees more specific comments and suggestions:

1. *The authors two-local system whose ground state implements the reversible Toffoli gate uses an ancilla bit. This bit's state is different for different inputs and outputs of the ground state. It is not clear how to interpret this. If the ancilla is included in the input our output, then the circuit is not reversible.*

On the question of the ancilla (the center spin) required to represent the Toffoli gate: this spin (labeled S in Fig. 1) is not part of the input or output spins (a, a', b, b', c , and d) to the gate, it is just an auxiliary degree of freedom. In the low energy sector that enforces the Toffoli gate (see the first eight entries of table I), the value of the spin S is fully determined by that of the other spins ($a = a', b = b', c$, and d). The reversibility concerns the spins a, a', b, b', c , and d .

2. *The authors should reference and compare their results to those in "Ground State Spin Logic" EPL 99 (2012) 57004 by J. D. Whitfield, M. Faccin, J. D. Biamonte.*

We thank the referee for pointing out the interesting paper by Whitfield, Faccin, and Biamonte. In the revised manuscript we referred to this as well as another 2008 paper by Biamonte (Ref. 17), together with the paper by Crosson *et al.* on ground state computing (Ref. 18).

3. *p1. "As we argue in more detail below, this should allow us to tackle a subclass of the Circuit Satisfiability (CSAT) problem and to solve factoring problems by using multiplication circuits with polynomial depth". The word "tackle" is very vague, the authors should clearly state that their approach allows one to formulate these problems in terms of the behavior of their physical system, not that they can "tackle" the problem.*

The word "tackle" was removed from the revised version and the application of our approach to factoring has been clarified.

4. *p1. "Here we focus on reversible rather than irreversible computation in order to address problems with both fixed-input and mixed-boundary conditions on inputs and outputs, as explained above." The paper reference appears to have a finite temperature phase transition (when used with fault tolerant circuits). The authors should comment that while these systems both encode computation in their ground state, they appear to be different from a statistic physics perspective. (Note also that Ref 17 fixes output bits in ancilla input bits in their complexity theory results, so it is not clear that mixed boundary fixing is novel).*

In the revised version, we added a discussion about the fact that error correction is needed in Ref. 18, while in our scheme accurate computation without error correction is possible below moderate temperatures that scale only as the inverse of the logarithm of the system size, a consequence of the exponential scaling of the static correlation length with inverse temperature. While Ref. 18, says that the error correction scheme works below a *critical* temperature, and this is suggestive of a phase transition, we could not determine with certainty that there is or not a transition in their scheme, and hence we would like to avoid a comparison that is not directly relevant to how we proceed with the model of computation. We prefer to stress the importance of dynamics instead.

On the issue of fixing the output bit, indeed Ref. 18 uses this scheme in their connection between ground-state spin computing and Promise-MA complete problems. What differs between the two schemes, and that we stress in the paragraph before we cite the prior literature (Refs. 17, 18, 19), is that reversible circuits allow for back-and-forward flow of information between the two boundaries. The difficulty that we see with irreversible circuits is that we cannot construct a transfer matrix backward, and that a pinned spin at the output opens bifurcating trajectories as one propagates backwards. This is not to say that reversible gates miraculously lead to solutions, but we can imagine dynamical obstructions that are lessened once these bifurcations are removed. Therefore, as a scheme for annealing a system to solution, we believe our proposal is original. For these reasons we stand by our statement that the case of mixed boundary conditions is one of the novel implications of reversibility.

5. *p1, "intuitive ideas from statistical mechanics" This should probably be more specific: ideas whose intuition comes from the statistics of spatially local systems.*

We have specified in various places in the new version that it is intuition in statistical mechanical systems with local interactions that stimulated the vertex model approach.

6. *p2. "In particular, the resulting thermodynamic behavior is that of a paramagnet" This contrasts with Ref. 17 and this should be pointed out.*

Please see our comments in point 4 above.

7. *p5, 'We present the result below, and leave the detailed justification for the Appendix.' this justification should be made inline here in a condensed form as it is a key feature of the rest of the paper to understand the model.*

We are afraid that the implementation of the TOFFOLI gate is too technical and adding details in the main text would detract from the main messages of the paper. For this reason we believe it is best to keep the details in the Supplementary Information.

8. *p5, "the calculation can be done straightforwardly in polynomial time via the column- by-column transfer" Please define transfer.*

We have now changed the language in the paragraph referred by the referee, which should clarify the picture. The relevant sentences read: "...if the N -bit input is fully specified and one is interested in the output, all that is needed is to transfer the information encoded into the input, left-to-right, by applying sequentially the gates one column of tiles at a time. In this case, if the depth (i.e., the number of steps) of the computation is polynomial in N , then this column-by-column computation reaches the output boundary, and thus solves the problem, in polynomial time."

9. *p7, "Any circuit can be laid down using these four rectangular tiles plus the TOFFOLI tile" Please use more precise terms that 'laid down'.*

We hope that the pictures in Figs. 1 and 2 in the revised version adequately illustrate the notion of "laying down the tiles" on a plane.

10. *p8. "Therefore, there are no phase transitions and the thermodynamics alone cannot reveal the complexity of the ground-state calculation, as it does not distinguish trivial problems (e.g., when all gates are ID or SWAP gates) from complicated ones (e.g., when the circuit contains a finite concentration of TOFFOLI gates)." It would seem that the more important argument is that this doesn't depend on the boundary conditions, since*

even TOFOLI gates in the forward direction can do what we consider traditionally not complex computations.

Indeed, the thermodynamics is independent of the circuit (and hence the concentration of TOFFOLIs) and of boundary conditions. (We note that, to stress that the dynamics does depend on the concentration of TOFFOLIs, we revised Fig. 5 and improved the discussion in the text.)

11. *p11, m was already used, please don't reuse variable names.*

The m counting the number of gate states has now been replaced with r .

12. *p15. "The addition of TOFFOLI gates complicates the analysis, but we find any scenario in which the phase transition disappears unreasonable" This seems completely unsubstantiated, one should not interject personal opinion unless clearly delineating that you are speculating.*

We have followed the referee's recommendation and changed the sentence to the following explanation: "The addition of TOFFOLI gates complicates the analysis, but on physical grounds we expect that the phase transition cannot simply disappear, but change character, from second order to first order. This could happen if the no-TOFFOLI critical point happens to be an endpoint of a phase boundary in the $\delta - c$ plane, where c is the concentration of TOFFOLI gates. However, determining the order of the transition for the vertex model describing a generic computation is a difficult problem, which we expect to address via quantum Monte Carlo simulations in a future publication. The conjectured equilibrium phase diagram for the quantum vertex model is shown in Fig. 10".

We thank the referee again for providing constructive criticism that helped us improve the manuscript.

REPLY TO REFEREE 3 – NCOMMS-16-08852

The paper "Solving classical computational problems by annealing a planar quantum vertex model" by Chamon et al. is a nice piece of work which provides a very interesting statistical mechanical view of quantum annealing. The introduction of an effective quantum vertex model allows for some detailed analysis of simulated annealing models. Given that their results are novel, I would only ask the authors to couch their results more deeply into the literature and clarify crucial points that were difficult to follow in the manuscript. In what follows, I will point out two places that can be clarified, give suggested references that may further inspire the authors, and finally give miscellaneous improvements.

We very much appreciate the referee's very positive comments. Below, we address the points he/she raised, and his/her suggestions.

1. *First, the paper takes a very nice approach inspired by a rich understanding of the vertex model. Hence, equation (8) is critical to understanding the key results of the paper. Thus, it is prudent to elaborate on equation (8) further. The symbols introduced in (8) are not well defined and some of them are unused in the remainder of the paper. Specifically, the term $K_{q_s, q'_s}^{g_s g'_s}$ is very confusing since there is already a variable K labelling the "grout" coupling. Also in (8) you have Δ_{q_s, q'_s} which you say for simplicity can be set uniformly but never have an example where the subscripts are used. A graphical representation e.g. over Figure 8, would be most helpful.*

Following the referee's suggestion, in order to clarify how the coupling tensor is constructed, we added an explicit example that we present in the Supplementary Information. There, it is clear that the tensor elements depend on the scalar coupling constant K . For the matrix Δ_{q_s, q'_s} , we choose to present the most general case in Eq. 8, but we only focus on the case where $\Delta_{q_s, q'_s} = \Delta$ independently of the vertex states. We hope that the explicit example for the K -tensor in the Supplementary Information clarifies our construction.

2. *Second, the authors try to relay the implications that their work has in the context of computer science complexity theory. Let me note that among NP problems not in P there are few are not proven to be NP-complete. These so called NP-intermediate problems include factoring and graph isomorphism. On page 2 in the sentence "When viewed from a computer science" as well as later in the paragraph containing eq (27) on pg 9,*

the authors compare their problems to graph isomorphism. While the authors have not made any undue claims, I urge the authors to take even more caution given that their paper didn't draw a clear line between the NP-complete and suspected NP-intermediate problems. Also on page 15 in the conclusion, I would suggest weakening their claim that "problems expected to be NP-complete may be solvable in quasi-polynomial time."

Regarding the implications of our work in the context of computer science complexity theory, we followed the referee's suggestion to weaken any claim that "problems expected to be NP-complete may be solvable in quasi-polynomial time." In the revised manuscript, we removed any claim about scaling of time-to-solution. The reason is that (as we also replied to referee 2), while the collapse of the data onto universal curves is *nontrivial*, our system sizes are not sufficient to conclusively extract time-to-solution as function of system size. To get the proper scaling of time-to-solution with system size would require large-scale simulations, in a scale beyond our present computational resources. We thus removed from the manuscript any unsubstantiated reference to the scaling of time-to-solution. As a consequence, we also removed the reference to the graph isomorphism problem.

We remark that, taking into account the comment by the referee, we made an effort to place the problems that we study in this manuscript within the proper complexity class. We note that we now stress in the text the difference between typical and worst-case scenarios: the problems that we study are drawn at random from a statistical ensemble of circuits. We also tried to frame the discussion on why we could reach solution in problems with a single solution, while our methods did not converge in problems with multiple solutions. (From a physics perspective, the problems with a unique solution correspond to non-competing ground states.) We believe this finding is again related to typicality, because solving the worst-case problems with a single solution is as hard as UNIQUE-SAT, which is proven to be as hard as SAT. Gaining a better understanding as to why the problems with unique ground state appear simpler to our methods, and clarifying the role of typical vs. worst-case scenarios in such problems, is an interesting problem; our work is just a first step in this direction.

3. *Concerning missing references, I think the paper has missed at least three key references. Most importantly, Nonperturbative k-body to two-body commuting conversion Hamiltonians and embedding problem instances into Ising spins by Biamonte. Physical Review A 77, 052331 (2008) is an earlier than the Crosson paper [17] and is aimed at the task of ground state encodings of classical problems. There was also a follow up by Biamonte and co-authors, Ground-state spin logic by Whitfield, Faccin, Biamonte. Europhysics Letters, vol 99, pg 57004 (2012) that is also relevant to the present article as it also gives constructions similar to those found in section II.A.*

We thank the referee for pointing out the relevant references to the work by Biamonte and collaborators, which we now cite (Refs. 17 and 19) in the introduction. We also cite Ref. 17 in section II, where we construct the logic gates using spins.

4. *Finally, there is an early reference on factoring as an optimization problem that contains a more detailed discussion that found on page 5 of your manuscript: Factoring as optimization by Burges. Microsoft Research MSR-TR-2002-83, Technical Report (2002) <https://www.microsoft.com/en-us/research/wp-content/uploads/2016/02/tr-2002-83.pdf>*

With respect to the Burges' Microsoft Research Report, "Factoring as Optimization," even though in the revised version we clarify more explicitly how using the vertex model representation of the multiplication circuit with mixed boundary conditions (i.e., in both the input and the output) can be used for the factorization of semi-primes, we do not attempt to discuss annealing of a factorization problem. We are currently working on a more elegant and efficient implementation of classical computations in the vertex model for that purpose, which will be presented in a future publication that will certainly cite Burges' report.

Miscellaneous comments: 1. On page, 1st paragraph "implies means" -> "implies" 2. In equations (21) it seems as though 'e' is missing if I use eq (19). This means (22) and (23) are also mistyped.

Finally, we addressed the miscellaneous comments that the referee made.

Once again we thank the referee for useful advice that helped us improve the manuscript.

Reviewer #1 (Remarks to the Author):

Unfortunately the authors still do not provide any evidence that quantum annealing for this particular problem has any advantage over other methods. Just claiming that the approach is good because it is novel is in my opinion insufficient.

Reviewer #2 (Remarks to the Author):

In response to the referees the authors have made many changes to the manuscript.

The paper is much improved by a reworking of the order and better clarity between which model is being discussed.

There are still places where I think the authors should be considerably more careful in their claims (listed below), however I do believe the results are interesting enough and of enough novelty to merit publication in Nature Communications.

Remaining issues the authors should optionally consider:

1) The authors make the claim on page 1 that in contrast to Ref [18] the model in this paper "is possible below moderate temperature that scale only as the inverse of the logarithm in the system size". It is not at all clear that this is actually better than the result in [18]. In [18] there is a finite temperature T below which errors exponentially are suppressed in the system size (amount of error correction added). In contrast the present work has the requirement of lowering a temperature to achieve error free computation. This effectively means the error correction has been offloaded to the system that is cooling more as a function of system size. It is not at all obvious how to efficiently achieve this in a physical system. It might be preferable if the authors simply contrast these results as two different approaches and do not make a direct comparison.

2) In their analysis of the perturbation of the transverse field on the system, I believe the authors are too optimistic about the power of degenerate perturbation theory. In particular the claim is that for $\lambda \ll J$ degenerate perturbation theory will only act on the local SAT manifold. While it is easy to rigorously prove this for a single transverse term, the errors in perturbation theory for each of these terms will scale like only one power less than perturbation parameter. If one then tries to do this N times, these errors add, and so one ends up requiring that N has to be less than one over the perturbation parameter or else one cannot avoid the UNSAT states becoming involved. In other words the energy scales must be adjusted such that they are only perturbative like one over the system size, this is a physical parameter whose precision must scale inversely like a system size. This does not seem physically achievable or 'fault-tolerant' as a computational device.

(For example of the kind of error analysis that leads to these conclusions see Phys. Rev. A 77, 062329 (2008) or Ann. Phys. Vol. 326, No. 10, pp. 2793-2826 (2011).)

Reviewer #3 (Remarks to the Author):

The changes are sufficient and I recommend the article for publication.

1) The authors make the claim on page 1 that in contrast to Ref [18] the model in this paper "is possible below moderate temperature that scale only as the inverse of the logarithm in the system size". It is not at all clear that this is actually better than the result in [18]. In [18] there is a finite temperature T below which errors exponentially are suppressed in the system size (amount of error correction added). In contrast the present work has the requirement of lowering a temperature to achieve error free computation. This effectively means the error correction has been offloaded to the system that is cooling more as a function of system size. It is not at all obvious how to efficiently achieve this in a physical system. It might be preferable if the authors simply contrast these results as two different approaches and do not make a direct comparison.

Indeed, we agree with the referee that these are two different approaches, each with its own characteristics. We stress that, in our approach, the system sizes for which we can compute error free grow exponentially in the ratio $K/2T$. Hence, temperatures on the scale of no less than one hundredth of K would allow us to compute error free in circuits of depth greater than 10^{20} . Therefore both our scheme and that discussed by the referee (utilizing error correction) are effective. We hope that our discussion in the introduction makes this point clear.

2) In their analysis of the perturbation of the transverse field on the system, I believe the authors are too optimistic about the power of degenerate perturbation theory. In particular the claim is that for $\lambda \ll J$ degenerate perturbation theory will only act on the local SAT manifold. While it is easy to rigorously prove this for a single transverse term, the errors in perturbation theory for each of these terms will scale like only one power less than perturbation parameter. If one then tries to do this N times, these errors add, and so one ends up requiring that N has to be less than one over the perturbation parameter or else one cannot avoid the UNSAT states becoming involved. In other words the energy scales must be adjusted such that they are only perturbative like one over the system size, this is a physical parameter whose precision must scale inversely like a system size. This does not seem physically achievable or 'fault-tolerant' as a computational device.

In the current version of the manuscript, there are two Hamiltonians, Eq. (1) and Eq. (11). The former acts solely on the projected Hilbert space of satisfied gates, whereas, in the latter, violations of the gate constraints are penalized by an energy scale $J \gg \Gamma, K, T$. Whether, in the limit of large J compared to K and Γ , the phase diagram topology is qualitatively the same or different for the Hamiltonians in Eq. (1) and (11) is the essential question. Along the classical path, the main focus of the current manuscript, the UNSAT manifold is exponentially suppressed in the ratio J/T . Hence, Hamiltonians (1) and (11) are equivalent along this path. A comparison of the behavior of the two Hamiltonians along the quantum path remains to be determined. We plan to address this point in a future

publication on Quantum Monte Carlo studies of the vertex model.